# Epigenetic Modulation of GPER Expression in Gastric and Colonic Smooth Muscle of Male and Female Non-Obese Diabetic (NOD) Mice: Insights into H3K4me3 and H3K27ac Modifications

**DOI:** 10.3390/ijms25105260

**Published:** 2024-05-11

**Authors:** Juanita C. Hixon, Jatna I. Rivas Zarete, Jason White, Mariline Hilaire, Aliyu Muhammad, Abdurrahman Pharmacy Yusuf, Benjamin Adu-Addai, Clayton C. Yates, Sunila Mahavadi

**Affiliations:** 1Center for Cancer Research, Department of Biology, Tuskegee University, Tuskegee, AL 36088, USA; jhixon@tuskegee.edu (J.C.H.); jawhite@msm.edu (J.W.); amachida31@gmail.com (A.M.); 2Department of Biomedical Sciences, College of Veterinary Medicine, Tuskegee University, Tuskegee, AL 36088, USA; jrivas4108@tuskegee.edu (J.I.R.Z.); baduaddai@tuskegee.edu (B.A.-A.); 3Department of Environment & Nutrition Sciences, College of Agriculture, Tuskegee University, Tuskegee, AL 36088, USA; mhilaire6334@tuskegee.edu; 4Department of Biochemistry, Faculty of Life Sciences, Ahmadu Bello University, P.M.B. 1044, Zaria 810211, Kaduna State, Nigeria; 5Department of Biochemistry, Federal University of Technology, P.M.B. 65, Minna 920101, Niger State, Nigeria; abdurrahmanpharmacy@futminna.edu.ng; 6Sidney Kimmel Comprehensive Cancer Center, Johns Hopkins School of Medicine, Johns Hopkins University, Baltimore, MD 21218, USA; cyates10@jhmi.edu

**Keywords:** type 1 diabetes, DNA methylation, histone modifications, smooth muscle, GPER, gender disparity, H3K4me3, H3K27ac

## Abstract

Type 1 diabetes (T1D) affects gastrointestinal (GI) motility, favoring gastroparesis, constipation, and fecal incontinence, which are more prevalent in women. The mechanisms are unknown. Given the G-protein-coupled estrogen receptor’s (GPER) role in GI motility, we investigated sex-related diabetes-induced epigenetic changes in GPER. We assessed GPER mRNA and protein expression levels using qPCR and Western blot analyses, and quantified the changes in nuclear DNA methyltransferases and histone modifications (H3K4me3, H3Ac, and H3K27Ac) by ELISA kits. Targeted bisulfite and chromatin immunoprecipitation assays were used to evaluate DNA methylation and histone modifications around the GPER promoter by chromatin immunoprecipitation assays in gastric and colonic smooth muscle tissues of male and female control (CTR) and non-obese diabetic (NOD) mice. GPER expression was downregulated in NOD, with sex-dependent variations. In the gastric smooth muscle, not in colonic smooth muscle, downregulation coincided with differences in methylation ratios between regions 1 and 2 of the GPER promoter of NOD. DNA methylation was higher in NOD male colonic smooth muscle than in NOD females. H3K4me3 and H3ac enrichment decreased in NOD gastric smooth muscle. H3K4me3 levels diminished in the colonic smooth muscle of NOD. H3K27ac levels were unaffected, but enrichment decreased in NOD male gastric smooth muscle; however, it increased in the NOD male colonic smooth muscle and decreased in the female NOD colonic smooth muscle. Male NOD colonic smooth muscle exhibited decreased H3K27ac levels, not female, whereas female NOD colonic smooth muscle demonstrated diminished enrichment of H3ac at the GPER promoter, contrary to male NOD. Sex-specific epigenetic mechanisms contribute to T1D-mediated suppression of GPER expression in the GI tract. These insights advance our understanding of T1D complications and suggest promising avenues for targeted therapeutic interventions.

## 1. Introduction

Diabetes mellitus (DM) is the leading cause of morbidity and mortality worldwide [1]. Current statistics of The International Diabetes Federation (IDF) have reported approximately 537 million cases and over 6.7 million deaths globally, of which type 1 diabetes (T1D) and type 2 diabetes (T2D) account for 10% and 90% of the cases, respectively [2]. However, despite its relatively low global prevalence, T1D is still a force to be reckoned with, and its burden increases annually, particularly among children, adolescents, and young adults [3]. Recent data suggest that North America is among the regions with the highest prevalence of T1D, while the lowest prevalence is seen in low- and middle-income countries (LMICs) such as East and West Africa, although the prevalence is not well reported in these regions [3].

Gastrointestinal (GI) complications, including heartburn, abdominal pain, nausea, vomiting, gastrointestinal reflux disease (GERD), gastroparesis, diarrhea, constipation, and fecal incontinence, have been reported in DM including T1D [4,5,6]. These complications are strongly correlated with poor glycemic control and commonly occur because of impaired gastrointestinal motility (GIM), which is mostly associated with diabetic autonomic neuropathy (gastrointestinal enteropathy) and abnormalities or the loss of Interstitial Cells of Cajal (ICC), the pacemaker of gastric smooth muscle [5,7]. Notably, signaling events initiated by the ligand-mediated activation of membrane-bound G-protein coupled receptors (GPCRs) also play critical roles in maintaining the normal contraction and relaxation of the tunica muscularis (the layer of smooth muscles lining the GI) to achieve a healthy GIM [8]. Therefore, altered expression or the inactivation of these receptors under diabetic conditions may lead to impaired GIM and GI complications in DM patients.

The role of G-protein-coupled estrogen receptors (GPERs) in the management of DM and its complications has recently emerged [9]. Indeed, estrogen-mediated GPER signaling confers additional protection against DM and its vascular complications in females who are yet to reach menopause compared to males and postmenopausal females [10,11]. This is consistent with the fact that GPER expression in some rodent and human tissues, including pancreatic β-cells, is higher in females than males [12,13]. Ironically, although sex disparities in the burden of GI complications of DM are not well documented, a few studies have reported a higher prevalence of GI complications (particularly gastroparesis) in women than in men with both T1D and T2D [14,15], despite the higher susceptibility of males to diabetes than females. This observation is quite mysterious, and thus, we hypothesize that a detailed understanding of GPER expression and signaling in the GI tissues of diabetic models could help unveil the mystery.

Evidence suggests that GPER activation enhances vascular smooth muscle relaxation and reverses diabetes-induced vasoconstriction in rodent models [16]. Similarly, GPER signaling inhibits colonic smooth muscle contraction and is involved in the regulation of GIM [17]. Therefore, impaired GPER expression or signaling can hinder GI smooth muscle relaxation, leading to hypercontractility and impaired GIM. However, to the best of our knowledge, there is currently no published research on GPER expression in GI smooth muscles using a T1D model. Hence, considering the potential role of GPER signaling in the regulation of GIM, we surmised that GPER expression may be altered under T1D conditions. Hence, we employed non-obese diabetic (NOD) mice, a popular murine model of T1D, to evaluate potential changes in GPER expression in gastric and colonic smooth muscle tissues. Similarly, we used both male and female mice to elucidate the potential sex-driven disparities in GPER expression under normal and diabetic conditions.

Epigenetic modifications, including changes in DNA methylation and histone modification patterns of chromatin and microRNA (miR)-mediated gene silencing, are reversible chemical changes that regulate gene expression without altering the DNA sequence [18]. DNA hypermethylation in the promoter region and the increased enrichment of repressive histone marks such as H3K27me3, with a concomitant decrease in active histone marks, such as H3K4me3 and H3K27ac, around the promoter regions of targeted genes are associated with gene silencing [19,20]. By contrast, gene activation involves promoter hypomethylation, the increased enrichment of active histone marks, and the decreased enrichment of repressive marks around the promoter regions of affected genes [20,21]. Here, we determined whether epigenetic mechanisms were involved in the regulation of GPER expression in the smooth muscle of the stomach and colon of NOD mice by examining DNA methylation and histone modifications (H3K4me3, H3ac, and H3K27ac) in the chromatin region harboring the GPER promoter. Our findings suggest that GPER expression is downregulated in the gastric and colonic smooth muscle tissues of NOD mice and that epigenetic mechanisms are involved in its downregulation.

## 2. Results

**Downregulation of GPER Expression in the Stomach and Colonic Smooth Muscle of NOD Mice:** The GPER expression is a pivotal factor in the pathogenesis of various diseases. To investigate potential variations in GPER expression between CTR and NOD mice, we quantified GPER expression at the transcriptomic level. Total mRNA was isolated from the gastric and colonic smooth muscle tissue strips and subjected to quantitative real-time PCR (qRT-PCR).

The GPER expression was significantly lower in the gastric smooth muscle of NOD mice when compared to the CTR of the same sex (Figure 1A–F). This difference was measured using three endogenous controls as reference genes. The statistical significance of these differences varied by each endogenous control, from *p* < 0.05 (GAPDH, Figure 1A) to *p* < 0.0001 (18S rRNA and β-actin; Figure 1C,E correspondingly) for males. For females, the size of the difference in GPER mRNA expression was consistently higher in CTR when compared to NOD mice (Figure 1B,D,F), being significant (*p* < 0.0001) with all endogenous controls (GAPDH, 18S rRNA, and β-actin).

The significance of the GPER mRNA expression levels between the sexes in CTR and NOD mice in the gastric smooth muscle varied, depending on the reference gene used (Figure 2A–F). Using GAPDH as a reference gene, GPER expression was significantly higher (*p* < 0.05) in female CTR and NOD mice than in their male counterparts (Figure 2A,B). However, this difference between the sexes disappeared (*p* > 0.05) in both NOD and CTR when 18S rRNA was the reference gene (Figure 2C,D). Finally, the difference (*p* < 0.05) reappeared when comparing male and female NOD mice, but not CTR (*p* > 0.05) when using β-actin as the reference gene (Figure 2E,F).

GPER expression was significantly lower in NOD mice’s colonic smooth muscle when compared to the CTR of the same sex (Figure 3A–F). The statistical significance of these differences varied by each endogenous control, from *p* < 0.001 (GAPDH and β-actin, Figure 3A,E, correspondingly) to *p* < 0.0001 (18S rRNA, Figure 3C) for males. For females, the size of the difference in GPER mRNA expression was also consistently higher in CTR when compared to NOD mice (Figure 3B,D,F), the difference being similarly significant (*p* < 0.01) with all endogenous controls (18S rRNA, GAPDH, and β-actin).

In the colonic smooth muscle, the GPER mRNA expression was compared between CTR and NOD male and female mice (Figure 4A–F). Using GAPDH or 18S rRNA as reference genes, female CTR mice had significantly higher GPER mRNA expression (*p* < 0.05) than males (Figure 4A,C). Similarly, female NOD mice have significantly higher GPER mRNA expression (*p* < 0.01 for GAPDH or *p* < 0.001 of 18S rRNA as references) than males (Figure 4B,D). The difference did not appear for CTR or NOD when it used β-actin as a reference gene (*p* > 0.05) (Figure 4E,F).

Consistent with the mRNA findings, GPER protein expression was lower in NOD male and female mice than in their CTR counterparts, both in the gastric (Figure 5A,B, Appendix A) and colonic smooth muscle (Figure 5C,D, Appendix A).

**Methylation ratios in the GPER promoter in the CTR and NOD gastric and colonic smooth muscle:** To determine the molecular mechanism underlying the downregulation of GPER in gastric and colonic smooth muscle, we first examined the CpG (cytosine–phosphate–guanine) sites found in the promoter of the GPER gene. We performed thorough sequencing of nine different genomic regions (identified as R1–R9) inside the GPER promoter sequence.

We examined the methylation status on R1 to R9 regions by targeted bisulfite methylation sequencing of the GPER promoter region. Upon evaluating the gastric and colonic smooth muscle tissue, our observations revealed that, within both CTR and NOD males and females, R1 exhibited a higher methylation ratio than R2–R9 in gastric smooth muscle (Figure 6A,B) (Table 1(A,B)). In the colonic smooth muscle, R1 was significantly more methylated than R3–R9 in both CTR and NOD male and female mice, but not significantly more than R2 (Figure 6C,D) (Table 1(C,D)).

Comparisons of homologous regions R1–R9 yielded no differences between the gastric and colonic smooth muscles of CTR and NOD mice of the same sex (Figure 7A–D) (Table 2).

**Methylation ratios at the GPER promoter in male vs. female CTR and NOD gastric and colonic smooth muscle:** Once again, in the gastric smooth muscle tissues of male and female CTR and NOD mice, R1 exhibited a higher methylation ratio than R2–R9, regardless of sex (Figure 6; Table 1(A,B)). In CTR and NOD male and female colonic smooth muscle tissue, we found that R1 exhibited a higher methylation ratio compared to R3–R9, for both sexes (Figure 6; Table 1(C,D)). There was no statistically significant difference in methylation ratios in the colonic smooth muscle of R1 and R2 in the CTR and NOD of both sexes.

However, the methylation ratios of homologous regions R1–R9 yielded no differences between CTR and NOD mice of either sex in gastric and colonic smooth muscle (Figure 8A–C; Table 3). In male NOD mice, we found a significantly higher methylation ratio in the R2 of the male colonic smooth muscles than in their female counterparts, indicating a possible sex disparity in their GPER promoter methylation patterns (Figure 8D, Table 3).

**Comparison of DNMT levels in gastric and colonic smooth muscles of male and female CTR and NOD mice:** Given the observed variations in DNA methylation patterns of R2 in the colonic smooth muscles of NOD mice (male vs. female), we investigated the levels of DNMT3A (Figure 9A,B) and DNMT3B (Figure 9C,D) enzymes in both CTR and NOD mice by examining both gastric and colonic smooth muscle tissues. Our analysis revealed no statistically significant differences in DNMT3A levels between CTR and NOD mice of box sexes in either gastric or colonic smooth muscle (percent change in Gastric smooth muscle: CTR vs. NOD male 60.1 ± 70.22; CTR vs. NOD female −12.62 ± 20.59; percent change in colonic smooth muscle: CTR vs. NOD male 27.79 ± 27.07; CTR vs. NOD female 75.82 ± 33.71) (Figure 9A,B).

We also discovered that no distinction had emerged in the case of DNMT3B (percent change in gastric smooth muscle: CTR vs. NOD male −5.50 ± 17.27; CTR vs. NOD female −0.74 ± 9.25; CTR male vs. CTR female −5.17 ± 21.86; NOD male vs. NOD female −0.48 ± 10.80; percent change in colonic smooth muscle: CTR vs. NOD male 14.68 ± 14.34; CTR vs. NOD female 44.04 ± 16.82; CTR male vs. CTR female 4.70 ± 18.27; NOD male vs. NOD female 25.88 ± 17.62) (Figure 9C,D).

**Comparison of H3K4me3 and H3K27ac levels in gastric and colonic smooth muscle of male and female CTR and NOD mice:** The previously noted discovery of the downregulation of GPER expression in both male and female NOD mice, particularly in their gastric and colonic smooth muscles, raises inquiries into the role of epigenetic factors in modulating GPER expression.

To explore this, we examined and compared methylation and acetylation levels in the histones of CTR and NOD mice within the gastric and colonic smooth muscles, as these factors could potentially influence GPER expression. Specifically, we assessed the levels of H3K4me3 and H3K27ac, which are mostly abundant around the transcription start sites of transcriptionally active genes in active chromatin.

Notably, no significant difference was observed between H3K4me3 levels in the gastric smooth muscle of male and female CTR and NOD mice (percent change in gastric smooth muscle CTR vs. NOD male 7.30 ± 6.27; CTR vs. NOD female 5.03 ± 4.59; CTR male vs. female 3.4 ± 2.01; NOD male vs. female −2.73 ± 5.54) (Figure 10A). However, male CTR mice displayed markedly higher levels of H3K4me3 in colonic smooth muscle than their NOD counterparts (Figure 10B). These levels were notably higher than those observed in female CTR mice. In other comparisons, no statistically significant differences were identified when evaluating H3K4me3 levels between CTR and NOD mice (percent change in colonic smooth muscle CTR vs. NOD male −21.63 ± 3.22; CTR vs. NOD female 29.07 ± 13.41; CTR male vs. female −27.76 ± 6.76; NOD male vs. female 12.37 ± 6.48 (Figure 10A,B).

In the gastric smooth muscle, H3K27ac levels were not significantly different between CTR and NOD mice of the same sex (Figure 10C). However, NOD male mice exhibited significantly higher levels of H3K27ac in the gastric smooth muscle than their NOD female counterparts (percent change in gastric smooth muscle CTR vs. NOD male 33.71 ± 10.69; CTR vs. NOD female −52.52 ± 10.45; CTR male vs. female 57.98 ± 12.34; NOD male vs. female −46.78 ± 7.551) (Figure 10C).

In the colon, a statistically significant reduction in H3K27ac levels was observed in male NOD mice compared with that in CTR male mice (Figure 10D). In contrast, NOD female mice displayed significantly elevated H3K27ac levels in the colonic smooth muscle compared to CTR female mice (Figure 10D). Furthermore, CTR female mice exhibited significantly lower H3K27ac levels in the colonic smooth muscle than CTR males (percent change in colonic smooth muscle CTR vs. NOD male −36.84 ± 5.40; CTR vs. NOD female 94.21 ± 36.57; CTR male vs. female −54.04 ± 7.55; NOD male vs. female 30.57 ± 10.45) (Figure 10D).

**Sex-specific H3K4me3 Chromatin Modifications at the GPER Promoter Region in CTR and NOD Gastric and Colonic Smooth Muscle:** To assess the presence of targeted histone marks near the GPER promoter, we conducted a ChIP assay using isolated total histone proteins. We observed a significant reduction in the enrichment of the (histone trimethylation mark) H3K4me3 in the gastric tissues of both male and female NOD mice (Figure 11A). However, we observed no significant reduction in the enrichment of the H3K4me3 in the colonic tissues of male and female NOD mice compared with their female counterparts (Figure 11B).

**Sex-Specific H3ac Chromatin Modifications at the GPER Promoter Region in CTR and NOD Gastric and Colonic Smooth Muscle:** Similar to H3K4me3 enrichment, a notable reduction in the enrichment of the global histone acetylation mark (H3ac) was observed in the chromatin region encompassing the GPER promoter in both male and female NOD mice in gastric smooth muscle tissues, relative to the CTR group (Figure 12A). There was a significant increase in the enrichment of H3ac in male NOD colonic smooth muscle compared with that in male CTR colonic smooth muscle, and in female NOD colonic smooth muscle. In contrast, the opposite occurred in females (Figure 12B).

**Sex-specific H3K27ac chromatin modifications at the GPER promoter region in CTR and NOD gastric and colonic smooth muscle:** We also conducted a ChIP assay using isolated total histone proteins to assess the presence of targeted histone marks H3K27ac near the GPER promoter. Male NOD mice and CTR female mice exhibited a statistically significant reduction in the levels of H3K27ac marks within their gastric tissues compared to male CTR mice (Figure 13A).

However, an opposite trend was observed in the colon. The enrichment of the histone acetylation marker, H3K27ac, was significantly increased in male NOD colonic smooth muscle compared to male CTR (Figure 13B).

Additionally, male NOD colonic smooth muscles showed higher enrichment of H3K27ac than female NOD ones (Figure 13B). On the other hand, female NOD colonic smooth muscles showed lower enrichment of H3K27ac than female CTR ones.

## 3. Discussion

This study explored the molecular mechanisms through which GPER expression is related to T1D digestive symptoms. To the best of our knowledge, this study provides the first evidence that GPER expression is downregulated in the stomach and colon of T1D mice and that sex-specific epigenetic alterations are involved in this downregulation.

GPER mRNA and protein expression decreased in male and female NOD mice’s gastric and colonic smooth muscles. This suggests that diabetes may contribute to the downregulation of GPER. Consistent with our findings, a recent study reported the suppression of GPER mRNA expression in the brain and peripheral blood mononuclear cells of the offspring of streptozotocin-induced diabetic mice [22]. According to Jiao et al. 2023, downregulation can result from inherited epigenetic modifications induced by diabetes [22].

Previous studies found a relationship between GPER activation and colonic smooth muscle relaxation, which altered the frequency of fecal pellet deposition in C57BL6 mice [17,23] and CD1 mice [17]. Li et al. [23] and Zielińska et al. [17] found that inhibiting GPER using G15, reduced the colonic transit time. We proposed that diminishing GPER expression could be equivalent to inhibiting or preventing its activation and, thus, could be expected to prevent the relaxation of muscles that would otherwise occur. Therefore, the under expression of GPER in the context of T1D may help explain the GI dysmotility found in T1D patients. This finding is consistent with the GI dysmotility phenotype observed in patients with T1D and T2D.

As explained by Oey et al., and Zhang et al. [24,25], the promoter region methylation of genes varies between individuals, even in inbred strains. This methylation is not indicative of different variants of a gene being present, and yet they can result in visibly different phenotypes being expressed within the same strain. This is exemplified by mice with the agouti yellow gene, which present different fur coat colors depending on the DNA methylation of the inter-individual differentially methylated regions. Similarly, methylation ratios and patterns can vary within different tissues of the same animal.

DNA hypermethylation is usually associated with gene silencing [26,27]. Interestingly, we observed different methylation patterns in our regions of interest, with R1 having a higher methylation ratio than R2–R9 in the GPER promoter, whereas R1 and R2 had similar CpG counts in all groups (Appendix A). The methylation ratios in male and female CTR, and NOD gastric and colonic smooth muscle R1, were higher than those in male and female R2–R9. The methylation ratio was lower in R2–R9 than in R1 between male and female gastric tissues in the CTR and NOD, stomach, and colon. These results are novel and will help to further characterize the promoter region of GPER in CTR and NOD mice. However, the methylation patterns were quite similar (R1 > R2–R9) for all male and female NOD and control mice except in the colon of the NOD mice, where the R2 of the male was hypermethylated compared to the R2 of the female. This is consistent with the fact that the male NOD mice had lower GPER mRNA expression compared to females. The methylation ratio in R1 was significantly higher in the NOD male colons than in the NOD female colons. The functions of R1 and R2 are unknown. Still, their methylation ratios were higher than those of other regions, suggesting that there is an activity and potential regulatory function of significance for this gene in these regions.

DNMT3A and DNMT3B protein levels were not significantly different between males and females or the CTR and NOD mice in the gastric and colonic smooth muscles, which implies that the methylation of R2 observed in the colon of male NOD mice compared to females might have resulted from an increase in DNMTs to the region rather than the upregulation of their protein expression.

Considering the extant differences in methylation, it is possible that the absence of differences in the DNMT levels means that other regulatory mechanisms are dominating the differences in methylation.

Histone modifications also play important roles in transcription regulation because they regulate chromatin accessibility via transcription machinery and serve as a scaffold for chromatin-modifying proteins that either write or erase other epigenetic marks [28]. Two important histone lysine modifications (H3K27ac and H3K4me3) that serve as markers of active chromatin are among the most commonly investigated histone modifications [29,30]. These marks are found in abundance around the promoters, enhancers, or near the transcription start sites (TSSs) of transcriptionally active genes, whereas transcriptionally silent genes are characterized by a paucity of these marks [29,31]. It is noteworthy that the abundance of H3K27ac marks at a particular region of the chromatin may lead to the establishment of the H3K4me3 mark in the region, and the domains of both marks overlap largely around the promoter regions of the active chromatin [29,31,32].

H3K4me3 levels were not significantly different between males and females in CTR and NOD gastric smooth muscles. H3K4me3 levels were significantly lower in the NOD male colon compared to the CTR male colon, whereas female CTR colon H3K4me3 levels were significantly lower than in CTR male colonic smooth muscle. A pattern was observed by which increased H3K4me3 levels or enrichment positively correlated with, and possibly resulted in, higher GPER expression levels, while lower enrichment or H3K4me3 levels were related to low GPER expression. The only exception to this was in the female NOD colonic smooth muscle tissue, in which GPER expression seemed to be independent of H3K4me3 levels and enrichment. This explanation may be found in the estrus cycles of the animals that were not measured.

Male NOD gastric smooth muscle H3K27ac levels were significantly higher than those of female NOD gastric smooth muscle, which were similar to the control. Therefore, neither H3K27ac nor H3K4me3 levels explain the GPER expression levels in the gastric smooth muscles of male NOD mice. Similarly, both H3K4me3 and H3K27ac levels were significantly lower in the NOD male colonic smooth muscle than in CTR, resulting in a double-risk system for the reduction of GPER expression in the colons of male NOD mice.

H3K27ac levels were significantly lower in NOD female gastric smooth muscle than in NOD male gastric smooth muscle, indicating that the H3K27ac levels may help explain disparities in the GI outcomes of males and females with T1D. H3K27ac levels, however, were higher in the colonic smooth muscles of NOD female mice than in those of CTR, despite lower GPER expression. Differences in epigenetic factors in different organs may result in differentiated risk for subsequent constipation and gastroparesis. It also suggests that lower GPER expression is not explained by H3K4me3 or H3K27ac alone in the colon of female NOD mice.

Finally, male CTR colonic smooth muscle H3K27ac levels were significantly higher than those of female CTR colonic smooth muscle, which may reduce the risk of decreased GPER expression. However, this only describes the potential for increasing or decreasing global gene expression levels, and the actual outcomes may be explained by the actual enrichment levels of the regions explored by the ChIP assay.

In the gastric smooth muscle, H3K4me3 and H3ac enrichment in the promoter region of the GPER gene was decreased in NOD male and female mice compared to CTR, justifying the lower GPER expression levels in NOD mice. H3K27ac enrichment was significantly lower in male NOD gastric smooth muscle than in the CTR. These results suggest that, in male NOD gastric smooth muscle, decreasing GPER expression is mediated by H3K4me3 and H3K27ac, whereas in female NOD gastric smooth muscle, only specific lysine residues of H3K4me3 and total H3 acetylation (H3ac) play a role in decreasing GPER expression.

According to Zhao et al. [31], H3K27ac seems to favor or even potentially induce gene expression. The pattern described by Zhao et al. [31] begins with H3K27ac installation, is followed by H3K4me3 enrichment (whose role in gene expression is unclear), which culminates with gene expression. The potential role of H3K27ac enrichment in gene expression is not clear.

In the gastric and colonic smooth muscle tissues of CTR and NOD mice, H3K4me3 enrichment seems to hold a certain proportion to H3K27ac enrichment levels (Figure 11 and Figure 13). Indeed, in the gastric smooth muscle, the H3K4me3 enrichment levels are significantly different among the same sexes, and vary in similar patterns as their corresponding H3K27ac enrichment levels (Figure 11A and Figure 13A). However, this relationship does not fully seem to hold in the colonic smooth muscle of CTR and NOD mice of both sexes, where the enrichment levels of H3K27ac are even tenfold higher than in the gastric counterpart, but H3K4me3 levels seem to be completely suppressed (Figure 11B and Figure 13B). Moreover, H3K4me3 levels do not seem to correspond directly to actual GPER protein expression levels (Figure 5 and Figure 11B). This suggests that the relationship between H3K4me3 enrichment is not essential to GPER gene expression. The effect of H3K27ac enrichment on H3K4me3 enrichment remains elusive. It still seems peculiar to us that H3K4me3 levels were extremely low in the colon (Figure 11B). Since H3K4me3 is an indicator of the frequent transcription of a gene (EpiQuick^TM^ Global Methylation Kit, 2022), these levels seemed so low, but not in proportion to actual GPER protein expression (Figure 11B) if H3K4me3 enrichment levels were predictive of GPER expression levels. A difference between the mechanisms underlying GPER protein expression in the gastric and colonic smooth muscles can be inferred. This may explain corresponding differences in GPER protein expression levels within the same animal in these different organs and may help explain why gastroparesis is not always accompanied by constipation, or vice versa, in conditions that are accompanied by lower GPER protein expression levels in both organs of the GI tract [33].

A noteworthy observation is that GPER protein expression in females seems to vary depending on the animal’s cycle stage. Differences due to different physiological conditions may account for the presence or absence of statistically significant differences in GPER expression levels between females. Future studies may strive to use both age and cycle-stage-matched animals to get more insight into the differences.

In colonic tissue, a reduced proportion of H3K4me3-marked regions was observed in male NOD mice, but the enrichment of this mark around the GPER promoter did not differ significantly from that of the control in both males and females. Surprisingly, the proportion of H3K27ac marks decreased significantly in the colon of male NOD mice, but the GPER promoter coverage of both H3ac and H3K27ac enrichment increased significantly compared to the male controls, indicating that diabetes could induce histone deacetylation in other chromatin regions of the male colon but not around the GPER promoter.

However, the enrichment of the H3ac mark around the GPER promoter decreased significantly in the colon of NOD females compared to that in female CTR, which is consistent with the significant downregulation of GPER expression in NOD females. Together, the reduced enrichment of the H3ac marks with the reduced H3K27ac enrichment levels also found in NOD females may explain the observed reduced GPER expression.

Intriguingly, the colons of male NOD mice showed greater hypermethylation of the R2 of the GPER promoter than those of NOD females (Figure 8D), although male NOD mice had higher GPER promoter coverage of both H3ac and H3K27ac than NOD females (Figure 12B and Figure 13B). This means that DNA methylation could have overridden histone acetylation to suppress GPER expression in the colon of male NOD mice, as both males and females had downregulated GPER mRNA levels compared to the CTR, and the downregulation was even more pronounced in males. Moreover, it has been previously reported that DNA methyltransferases (DNMTs) must recognize and bind to unmethylated H3K4 through their ADD domain before adding methyl groups to the DNA [34,35]. This implies that the lack of H3K4me3 encourages DNMT binding, leading to increased hypermethylation observed in the R2 region of male NOD mice relative to females. Moreover, considering that no significant differences were observed in the protein content of DNMTs between male and female NOD mice, we may conclude that the observed hypermethylation in the colon of male NOD mice compared to that in females might result from the increased recruitment of DNMTs to the GPER promoter rather than an increase in their protein levels.

Our findings indicate that epigenetic changes in the promoter region of GPER not only correlate with expression levels consistent with the types of epigenetic modifications found (acetylation and methylation), but also that certain modifications are consistently found in mice with T1D.

The limitations of this study represent opportunities for further research and nuance in the interpretation of results. One limitation is that the stages of the estrus cycle of female mice have an impact on GPER expression and GI motility [33] and may have an impact on the exact epigenetic characteristics of the DNA and histones of our samples derived from female mice; however, these stages were not measured in the current study. Other factors that were not measured may also contribute to the GPER expression levels of the samples. These factors include histone deacetylation and the levels of corresponding enzymes, such as histone acetyltransferases and miRNAs. In addition, we only searched part of the promoter region of GPER, which was an area associated with H3 histones; changes in methylation in other areas or effects in other histones may have also occurred, and thus, would be unaccounted for in this study, even though it is possible that DNA methylation is region-specific. Whole genome sequencing may be necessary to account for all parts of the GPER gene.

Lastly, there is also limited information on the epigenetic modification of GPER expression. Overall, this study offers novel information regarding the significance of diabetes in GPER expression and regulation in uncharted territories.

## 4. Materials and Methods

**Animals:** Approximately 10–14 week-old age-matched male and female CTR (Control, NOD.B10Sn-H2b/J) and type I non-obese diabetic (NOD, NOD/ShiLtJ) mice were purchased from Jackson Laboratories and housed in a 12h/12h dark/light cycle and provided food and water ad libitum in the animal facility provided by the Division of Animal Resources, Tuskegee University. All procedures were approved by the Institutional Animal Care and Use Committee of Tuskegee University before the start of the experiments. For sampling, the matching/pairing between specific males and females was carried out randomly.

**Preparation of Gastric and Colonic Smooth Muscle Strips:** Eleven cohorts of four mice each (one male CTR, one female CTR, one male NOD, and one female NOD) were sacrificed throughout the study. The mice were euthanized by CO_2_ inhalation, followed by cervical dislocation. The stomach and colon were removed, and emptied of their contents, followed by flushing with PBS. Then, the gastric and colonic mucosa layers were scraped to separate them from the muscle layers of their corresponding organs, resulting in smooth muscle strips. Smooth muscle strips from gastric and colonic smooth muscle layers were used for genomic DNA extraction for bisulfite and chromatin immunoprecipitation (ChIP) assays, nuclear and total extractions for the DNMT3A/B assay, histone K4 trimethylation (H3K4me3), histone K27 acetylation (H3K27AC), RNA isolation, qPCR, and Western blot analyses [36].

The smooth muscle samples isolated from CTR and NOD mice sacrificed on the same day were processed together as matched pairs (CTR male vs. NOD male, CTR female vs. NOD female) to ensure identical ages for comparison. A total of 44 animals were thus sacrificed and pair-matched by the end of the study. Consequently, all experiments consisted of 44 biologically independent animals and were run in triplicate.

**Isolation of RNA from the gastric and colonic smooth muscles of male and female CTR and NOD mice:** Gastric and colonic smooth muscle samples from the CTR and NOD groups were subjected to a series of biochemical steps to extract RNA. Each sample was homogenized in 0.5 to 1 mL of TRIzol, and the resulting homogenate was transferred to a microfuge tube. To this, 0.2 mL of RNAse/DNAse-treated water and chloroform were added, and the mixture was incubated at room temperature for 15 min. After incubation, the mixture was centrifuged at 12,000× *g* for 15 min at 4 °C. The top aqueous phase, which contained the RNA of interest, was carefully separated and transferred into fresh tubes. Subsequently, isopropanol (0.5 mL of isopropanol) was added to the aqueous phase. The resulting solution was incubated at −80 °C overnight and then subjected to another round of centrifugation at 12,000× *g* for 15 min at 4 °C. After this step, the supernatant was discarded, leaving behind the pellet that was resuspended in 1 mL of cold ethanol (75%). After resuspension, the mixture was centrifuged again at 12,000× *g* for 15 min. The supernatant was removed and the remaining pellet was air-dried for 15 min at room temperature. The concentration of total RNA was determined by spectrophotometry after the RNA was dissolved in 0.1% diethylpyrocarbonate (DEPC)-treated water [33].

**Quantitative reverse-transcription PCR (qRT-PCR) analysis:** Following the RNA extraction process, each sample was subjected to reverse transcription using the High-Capacity cDNA Reverse Transcription kit in a 20 mL reaction volume. TaqMan PCR Master Mix was used to perform quantitative real-time PCR (qRT-PCR) on the cDNA samples. The samples corresponded to groups of 4–5 animals per each of the groups (CTR and NOD). The tests were run in triplicates per mouse. TaqMan-specific primers targeting GPER (Mm01194815_m1), β-actin (Mm02619580_g1), GAPDH (Mm05765180_g1), and 18S rRNA (Mm02601777_g1) were used in this study. The threshold cycle parameter, or the value at which the fluorescence generated by the probe cleavage was above a predetermined threshold, was used to determine the initial copy number of the target gene. To determine the relative expression of the genes (fold change), we followed the following procedure.

First, the cycle threshold (C_T_) of the GPER gene was normalized to the Ct of the reference genes (in this case, beta actin, GAPDH and 18S rRNA) for the CTR and NOD groups. Then, the resulting normalized C_T_ of GPER of NOD was compared to the normalized C_T_ of GPER of the CTR group (this is known as the 2^−ΔΔCT^ method) [37,38,39,40,41,42,43]. The final results were expressed as the fold difference in gene expression between the NOD and CTR mouse samples. All PCR were performed using ABI QuantStudio 5 or the 7500 Fast Real-Time PCR system [37].

**Western blot analysis:** We used a lysis buffer based on Triton X-100 containing protease and phosphatase inhibitors (100 g/mL PMSF, 10 g/mL aprotinin, 10 g/mL leupeptin, 30 mM sodium fluoride, and 3 mM sodium vanadate) to prepare gastric and colonic smooth muscle strips from both CTR and NOD mice; protein concentration in the supernatant was measured using a Bio-Rad Dc protein assay kit after the centrifugation of the lysates at 20,000× *g* for 10 min at 4 °C. Equal quantities of these proteins were separated by SDS-PAGE and transferred onto PVDF membranes. These membrane blots were subsequently blocked with a 5% (*w*/*v*) non-fat dried milk/TBS-T solution (Tris-buffered saline with pH 7.6 and 0.1% Tween-20) for one hour before they were incubated with antibodies, GPER (Genetex Inc., Irivine, CA, USA; 1:1000) or β-actin (Sigma-Aldrich Inc., St. Louis, MO, USA; 1:5000), in PBS-T with an additional 5% (*w*/*v*) non-fat dried milk. Following a one-hour incubation in PBS-T (phosphate-buffered saline with 0.1% Tween-20) containing 1% (*w*/*v*) non-fat dried milk with the corresponding horseradish peroxidase-conjugated secondary antibody (1:5000), immunoreactive proteins were visualized using the Pierce ECL Western blotting Substrate Kit (ThermoFisher Scientific, Waltham, MA, USA). Throughout the procedure, PBS-T was used for all washing steps, and the protein bands were identified using the enhanced chemiluminescence reagent within the GE Amersham 680 imaging system [36]. The quantitative analysis was conducted by measuring the band intensity using ImageJ software, ImageJ 1.54; Java 1.8.0_345.

**Targeted bisulfite sequencing of gastric and colonic smooth muscles from male and female CTR and NOD mice:** The bisulfite assay was used to analyze the methylation status of cytosine residues in a DNA molecule in the gastric and colonic smooth muscles of male and female CTR and NOD mice. MethylCheckTM Service was used to process and analyze the samples. Assays were designed to target CpG sites in the GPER promoter-specified region of interest (ROI) using primers designed using Zymo Research’s proprietary sodium bisulfite-converted DNA-specific primer design tool Rosefinch. Primers were designed to minimize amplification bias by synthesizing ordered primers with pyrimidine (C or T) at the CpG cytosine in the forward primer and purine (A or G) at the CpG cytosine in the reverse primer. The primers were validated using real-time PCR with bisulfite-converted CTR DNA, and the PCR products were confirmed by DNA melt analysis. After primer validation, the samples were bisulfite-converted using the EZ DNA Methylation-Lightning Kit (Zymo Research, D5030), following the manufacturer’s instructions. All samples were amplified using ROI-specific primer pairs, with limited amplification cycles. The resulting amplicons were pooled, barcoded, and purified using a DNA Clean & Concentrator-5TM kit. The samples were then prepared for massive parallel sequencing using an Illumina MiSeq V2 300 bp Reagent Kit and paired-end sequencing protocol, as directed by the manufacturer. Sequence reads were identified using standard Illumina Miseq Real-Time Analysis (RTA) software version RTA 1.18. and analyzed using a proprietary analysis pipeline written in Python by Zymo Research. The methylation level in each cytosine sample was calculated by dividing the number of reads containing C by the total number of reads containing C or T.
**GPER Promoter Region of Interest Sequence:**
**Small leteers = Repeat Sequences; Capital Letters = Unique Sequence; Dark Gray = Regions not covered by Targeted Sequencing Amplicons, Light Gray = Regions of Interest; CG = CG of interest**
>mm39_dna range=chr5:139408385–139409084 5′pad = 50 3′pad = 50 strand = + repeatMasking = lowerCCACCTTGGTTGTACACGGTCTAGAACTAGAGACAGAGCTGACCCCTGGGTAGAGGGACATTAGCCCTAGCTAACATTAGAGTAGAAAACACACCTGGATTCCTAATTTCTGGTCAAAATGCCGGCTACTTGTAGACTGTATTTCCCACTTCGAGCATCCCTAGGAAAACTATAGCTTACATTTGCTGTGTGACTGCAGTCCTACCATGCTGATTGAGGAAACTCAGTGTTCAGTTTTAAAGCCAGCACAAGCTAAAACAGGCAAGGGTATCATTGCTTCAACAAATGAGGAAGGATTCTTACCTAAAAGGTAAACAATATTATAATCCTTTCACACTTTAAATAAATGTGTCAGTGGGTGAAAATCAGCAGTCACACTGGAAACTTCCATAAAATACACATCCAGCAGGGTCGTTTTCACTGTCTACATGTGGGAGGAAAAAAACTGCCAGCAAAAAAATGGTTAATGCTGGCCTTAAAGGGAGGCTGGCCTTAAAGGGAGGCTGGCCAGAGCCCAGTGAGTAGGCTTGGGAAGTCTATAAAGGAGGCGCTGTGCCAAGGGGGCCAGACGCTGCTGGACGGCCACAGGCATCCATCCCCAGGCATCGGGCGGGTGCTTCTGTTCCTCTCCTGCTGGGTCCCTGCTGGGCACCGTCCCCAAAGTGCTGCAAGTCCAGGGTCCATCCCTGGAGCAAGCTCCAGGAGCACCTCCAGCAGEstimated number of amplicons: 3



**chromosome**

**Start**

**End**

**Region**

**chr5**

**139408506**

**139408507**

**1**

**chr5**

**139408535**

**139408536**

**2**

**chr5**

**139408796**

**139408797**

**3**

**chr5**

**139408915**

**139408916**

**4**

**chr5**

**139408936**

**139408937**

**5**

**chr5**

**139408946**

**139408947**

**6**

**chr5**

**139408973**

**139408974**

**7**

**chr5**

**139408977**

**139408978**

**8**

**chr5**

**139409019**

**139409020**

**9**



**Evaluation of DNA Methyl Transferase 3A and 3 B in the Gastric and Colonic Smooth Muscle of Male and Female Mice from both the CTR and NOD Groups:** The DNA Methyltransferase 3A & 3B Assay Kit (Epigentek, Brooklyn, NY, USA) was used in this analysis. Nuclear extracts were derived from gastric and colonic smooth muscle samples obtained from male and female CTR and NOD mice, respectively. Specifically, ten micrograms of nuclear extract were subjected to a series of steps, beginning with incubation with the capture reagent and assay buffer for 2 h at 37 °C. Following this incubation, the samples were exposed to the affinity antibody for 60 min and detection antibody for 30 min at room temperature. To quantify the results, absorbance was measured at 450 nm using a microplate spectrophotometer. The percentage change in DNA methyltransferase (DNMT) was calculated according to the following formula: [(treated (NOD) sample OD − blank OD)/(Untreated (CTR) sample OD − blank OD)] × 100, according to the manufacturer’s instructions. Protein standards of known concentrations (30, 20, 10, and 2 ng) were used to generate a standard curve. The final outcome was expressed as the percentage change in the DNMT3A and DNMT3B expression levels.

**Isolation of Nuclear Proteins from the Gastric and Colonic Smooth Muscle Tissues of both Male and Female Mice in the CTR and NOD Groups:** To obtain nuclear proteins, the EpiQuick Nuclear Extraction Kit (Epigentek) was used following the manufacturer’s guidelines. Approximately 20 mg of smooth muscle tissue from the stomach and colon was minced and homogenized in NE1 buffer using a Dounce homogenizer. After homogenization, samples were collected and centrifuged for 10 min at 12,000 rpm. The nuclear extract was prepared by adding two volumes of NE2 buffer to the resulting pellet, followed by sonication. This was followed by another centrifugation step at 14,000 rpm for ten minutes. The nuclear extracts were stored at −80 °C until further analysis. The QuickStart Bradford Protein Assay (Bio-Rad, Hercules, CA, USA) was used to determine the protein concentrations of the nuclear extracts.

**Extraction of Total Histones from the Gastric and Colonic Smooth Muscle of Male and Female Mice in both the CTR and NOD Groups:** To isolate total histones, the EpiQuick Total Histone Extraction Kit was used following the manufacturer’s instructions. Approximately 20 mg of smooth muscle tissue from the stomach and colon was finely minced and homogenized using a Dounce homogenizer in a pre-lysis buffer. Following homogenization, the prelysate mixture was centrifuged for one minute at 10,000 rpm. Total histone extraction was performed by adding three volumes of lysis buffer to the resulting pellet. The lysates were then centrifuged at 12,000 rpm for 5 min, and 0.3 volumes of balanced DTT buffer were introduced to the supernatant. The total histone extracts were preserved at −80 °C until further analysis. The QuickStart Bradford Protein Assay (Bio-Rad, Hercules, CA, USA) was used to assess the protein concentration of the nuclear extract.

**Assessment of Histone 3 Lysine 4 Trimethylation (H3K4Me3) and Histone 3 Lysine 27 Acetylation (H3K27AC) in Gastric and Colonic Smooth Muscle Tissues from Male and Female CTR and NOD Mice:** EpiQuick H3K4Me3 and H3K27A C assay kits were used to measure the amounts of Histone 3 Lysine 4 Trimethylation (H3K4Me3) and Histone 3 Lysine 27 Acetylation (H3K27AC). Total histone extracts from the stomach and colonic smooth muscle samples of male and female CTR and NOD mice were used for these tests. Briefly, the amounts of various histone modifications were measured using one microgram of histone proteins. Methylated and acetylated histones were detected using the corresponding specific antibodies using ELISA-based methods. They were then measured using a color development reagent and a detection antibody. Colorimetric measurements were performed at a 450 nm wavelength using a Promega GloMax absorbance plate reader. The percentages of H3K4Me3 and H3K27AC were calculated according to the following formula: [(treated (NOD) sample OD − blank OD)/(untreated (CTR) sample OD − blank OD)] × 100, in accordance with the manufacturer’s instructions. For reference and calibration, protein standards with known concentrations (ranging from 1.5 to 100 ng/µL) were incorporated to generate a standard curve. The results are presented as the percentages of H3K4Me3 and H3K27AC.

**Protein cross-linking and chromatin immunoprecipitation of male and female gastric and colonic smooth muscles from male and female CTR and NOD mice:** We performed a ChIP assay using a modified version of a protocol described previously [44]. Before the chromatin immunoprecipitation assay, the proteins were cross-linked to DNA following a specific protocol, with some modifications. Briefly, gastric and colonic smooth muscle strips were placed immediately in liquid nitrogen and stored at −80 °C until the next step. Gastric and colonic smooth muscle strips were cut into small pieces, placed in a 15-mL centrifuge tube containing DPBS (pH 7.4) with 1% formaldehyde, and kept at room temperature (RT) for 8–10 min. The reaction was stopped by adding 10× glycine at a final concentration of 0.125 M and 5 min incubation at RT.

The tissue was washed four times with cold phosphate-buffered saline (PBS) containing protease inhibitors (Complete; Roche, Branchburg, NJ, USA) and homogenized in ice-cold cell lysis buffer (10 mM NaCl, 0.2% Nonidet p-40, and 10 mM Tris-HCl, pH 8.0) with protease inhibitor cocktail II. The nuclear fraction was lysed in ice-cold nuclear lysis buffer (10 mM EDTA, 1% SDS, and 50 mM Tris-HCl, pH 8.0) with protease inhibitor cocktail II and sonicated on ice using an active motif sonicator. The lysate was centrifuged to remove insoluble material and then diluted 1:10 in ChIP dilution buffer (167 mM NaCl, 0.01% SDS, 1.1% Triton X-100, 1.2 mM EDTA, and 16.7 mM Tris-HCl; pH 8.1) to a final volume of 1.0 mL. Primary antibodies were added to the diluted lysates and incubated with 15 mL of fully suspended protein A/G magnetic beads (ThermoFisher Scientific, 88802) at 4 °C for 12 h. The following primary antibodies were used: acetyl-histone H3 (EMD Millipore; 06-599, 1:200), acetyl-histone H3 (Lys27) (Active Motif; 39133, 10 mg), and tri-methyl-histone H3 (Lys4) (Active Motif; 39159, 1:200). The chromatin/immune complexes were washed with a low-salt immune complex wash buffer (150 mM NaCl, 0.1% SDS, 1% Triton X-100, 2 mM EDTA, and 20 mM Tris-HCl, pH 8.1), high-salt immune complex wash buffer (500 mM NaCl, 0.1% SDS, 1% Triton X-100, 2 mM EDTA, and 20 mM Tris-HCl, pH 8.1), LiCl immune complex wash buffer (0.25 M LiCl, 1% NP-40, 1% deoxycholic acid sodium salt, 1 mM EDTA, and10 mM Tris-HCl; pH 8.1), and TE buffer (1 mM EDTA and 10 mM Tris-HCl, pH 8.0). The chromatin/immune complexes and input DNAs were reverse cross-linked by incubation with ChIP elution buffer (1% SDS and 0.1 M NaHCO_3_) containing 0.1 mg/mL proteinase K for 2 h at 62 °C. DNA was purified by using spin columns (EMD Millipore). Input and immunoprecipitated DNAs were subjected to qPCR using the promoter region of the GPER primers F: 5′CCCAGTGAGTAGGCTTGGGAA 3′ and R: 5′AGCACTTTGGGGACGGTG 3′. All assays included non-immune IgG for CTR to determine the specificity of each antibody used. All reactions were confirmed to generate a single PCR product by gel melting curve analysis. Data are shown as the fold enrichment of IgG for male and female CTR versus NOD mice, depending on the antibodies tested. 

**Statistics and Reproducibility:** All data were subjected to rigorous statistical analysis. Two-way analysis of variance (ANOVA) with multiple comparison tests was used to analyze the results comprehensively. Specifically, mRNA expression data were analyzed using Student’s *t*-test, considering both paired and unpaired values. Statistical analyses and graphical representations were performed using GraphPad Prism 9.5.0(730) (GraphPad Software, Inc., Boston, MA, USA). To convey the findings, data are presented as the mean and standard error of the mean (SEM), with statistical significance set at *p* < 0.05. It is important to note that each statistical analysis was based on data obtained from at least three independent experiments. An experiment was considered independent when it was derived from distinct samples, thereby reflecting the biological replicates. Figure legends provide information on the sample size for each experiment. It is worth emphasizing that the results should be reproducible, provided that the same methods were followed, including using the same mouse strains, and that the mice were maintained under controlled environmental conditions.

## 5. Conclusions

GPER expression was lower in T1D NOD mice in both gastric and colonic smooth muscles of male and female mice. Mechanistically, the variations observed in the modification of the histones around the GPER promoter region may be partly responsible for the downregulation of GPER expression in the GI smooth muscles of the NOD mice compared to CTR mice. However, some other mechanisms may also help explain the differences in GPER expression between NOD and CTR mice. Further studies may help elucidate the exact pathway for GPER expression regulation in the GI tract of T1D patients to discover targets to treat or to prevent GI complications of this metabolic condition.

## Figures and Tables

**Figure 1 ijms-25-05260-f001:**
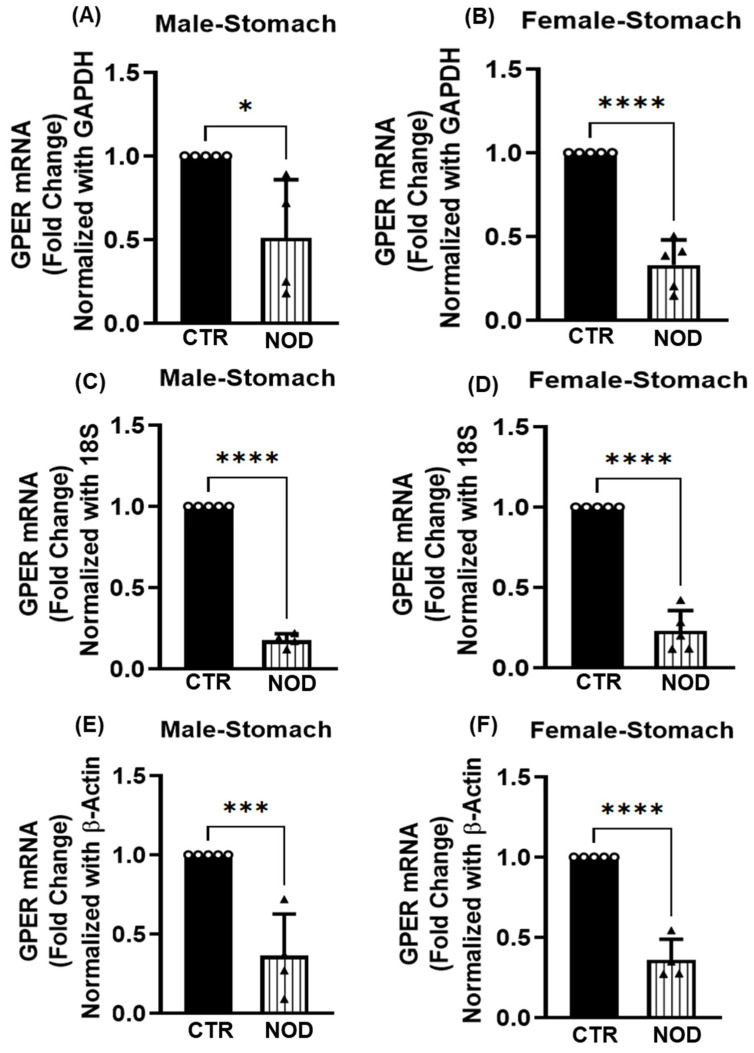
**GPER mRNA expression in CTR versus NOD male and female gastric smooth muscle tissues**. Total mRNA was isolated from gastric smooth muscle tissue strips and subjected to quantitative real-time PCR (qRT-PCR). In gastric smooth muscle tissues, both male and female CTR mice exhibited significantly elevated GPER mRNA expression levels compared to those of NOD mice. GAPDH (**A**,**B**), 18S rRNA (**C**,**D**), and β-Actin (**E**,**F**) were used as endogenous controls. Results were deemed significant when *p* < 0.05; *t*-test analysis *p*-value (* *p* = 0.015, **** *p* < 0.0001, GAPDH, 18S rRNA, *** *p* = 0.0009, **** *p* < 0.0001, β-Actin; *n* = 5). Open circle represents CTR and black triangle represents NOD samples.

**Figure 2 ijms-25-05260-f002:**
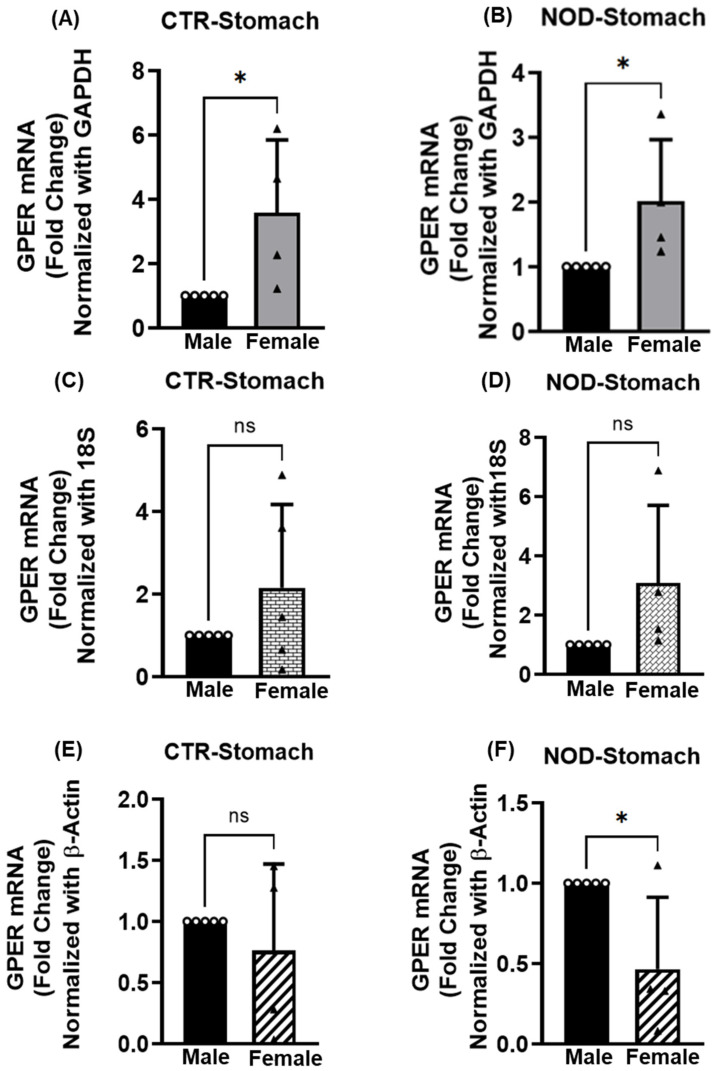
**GPER mRNA expression in male versus female CTR and NOD gastric smooth muscle tissues**. Total mRNA was isolated from gastric smooth muscle tissue strips and subjected to quantitative real-time PCR (qRT-PCR). In gastric smooth muscle tissues, female CTR and NOD mice exhibited elevated GPER mRNA expression levels compared to those of male CTR and NOD mice where GAPDH (**A**,**B**), 18S rRNA (**C**,**D**) were used as endogenous controls. Conversely, contrasting results were observed when β-Actin (**E**,**F**) was employed as an endogenous control. Results were deemed significant when *p* <0.05; *t*-test analysis *p*-value (* *p* = 0.0346, 0.0455, GAPDH; ns *p* > 0.05, 18S rRNA; ns *p* > 0.05, * *p* = 0.0298, β-Actin *n* = 5). Open circle represents male and black triangle represents female samples.

**Figure 3 ijms-25-05260-f003:**
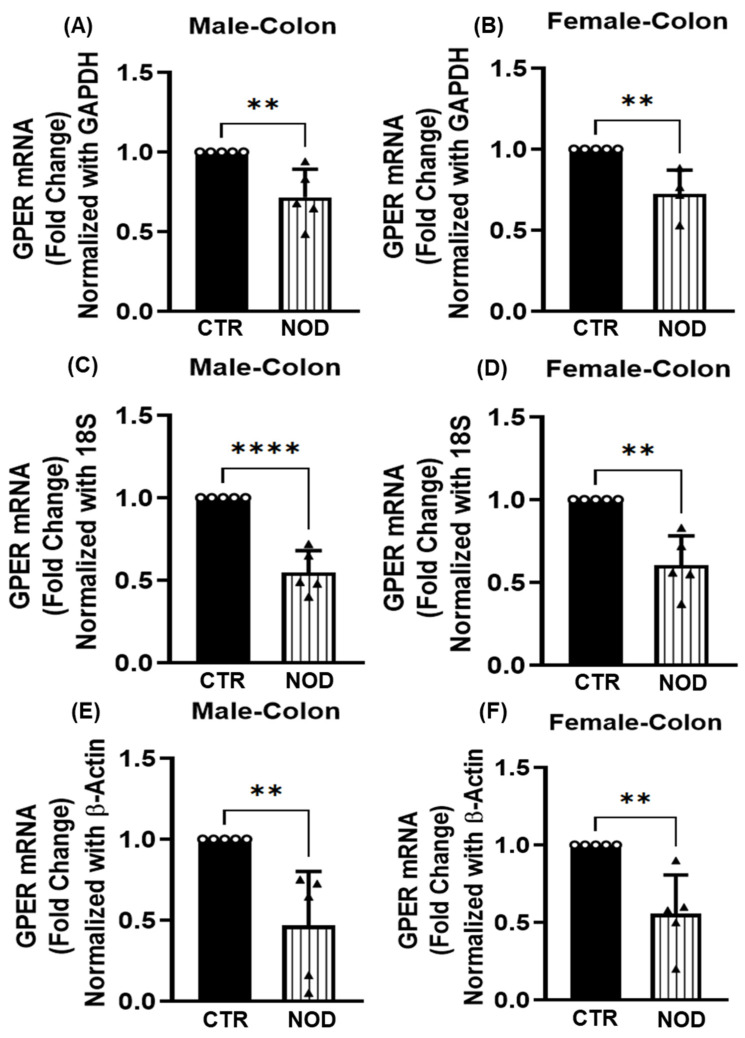
**GPER mRNA expression in CTR versus NOD colonic smooth muscle tissues**. Total mRNA was isolated from colonic smooth muscle tissue strips and subjected to quantitative real-time PCR (qRT-PCR). In colonic smooth muscle tissues, both male and female CTRl mice exhibited significantly elevated GPER mRNA expression levels compared to those of NOD mice. GAPDH (**A**,**B**), 18S rRNA (**C**,**D**), and β-Actin (**E**,**F**) were used as endogenous controls. Results were deemed significant when *p* <0.05; *t*-test analysis *p*-value (** *p* = 0.007, 0.0074, 0.0011, 0.0038, 0.0041; **** *p* < 0.0001; *n* = 5). Open circle represents CTR and black triangle represents NOD samples.

**Figure 4 ijms-25-05260-f004:**
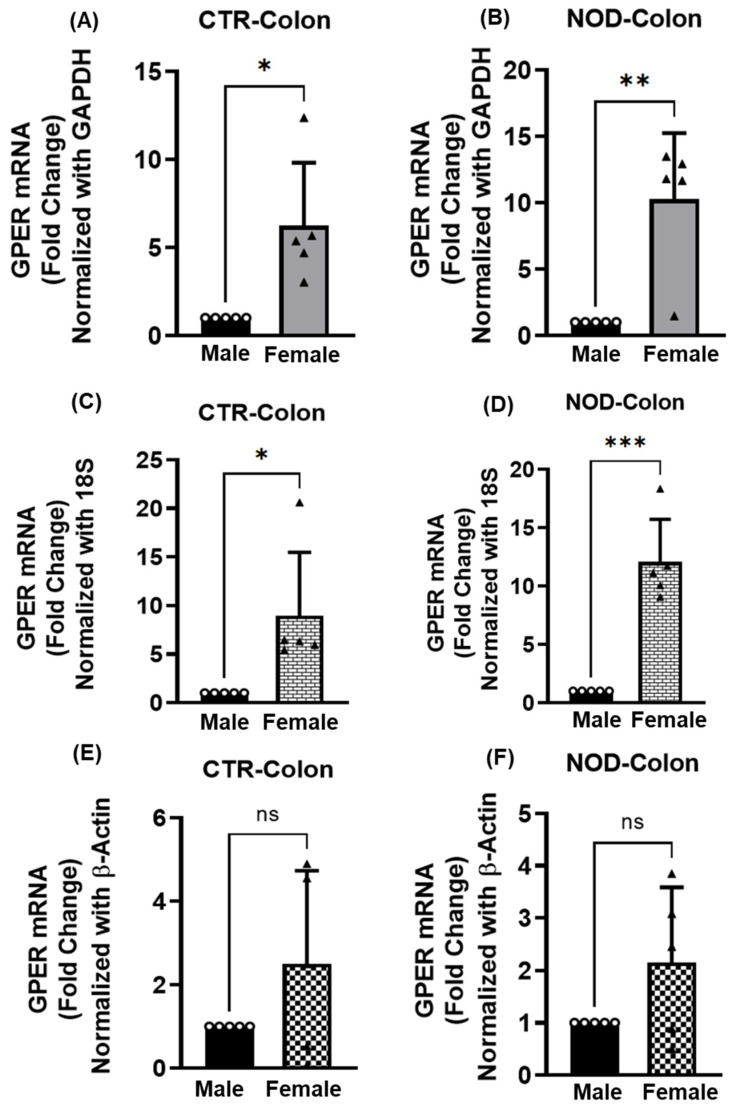
**GPER mRNA expression in male versus female CTR and NOD colonic smooth muscle tissues**. Total mRNA was isolated from colonic smooth muscle tissue strips and subjected to quantitative real-time PCR (qRT-PCR). In colonic smooth muscle tissues, female CTR and NOD mice exhibited significantly elevated GPER mRNA expression levels compared to those of male CTR and NOD mice. GAPDH (**A**,**B**), 18S rRNA (**C**,**D**), and β-Actin (**E**,**F**) were used as endogenous controls. Results were deemed significant when *p* <0.05; t-test analysis *p*-value (* *p* = 0.0113, ** *p* = 0.0031, GPADH; * *p* = 0.0262, *** *p* = 0.0001; 18S rRNA; ns *p* > 0.05, β-Actin; *n* = 5). Open circle represents male and black triangle represents female samples.

**Figure 5 ijms-25-05260-f005:**
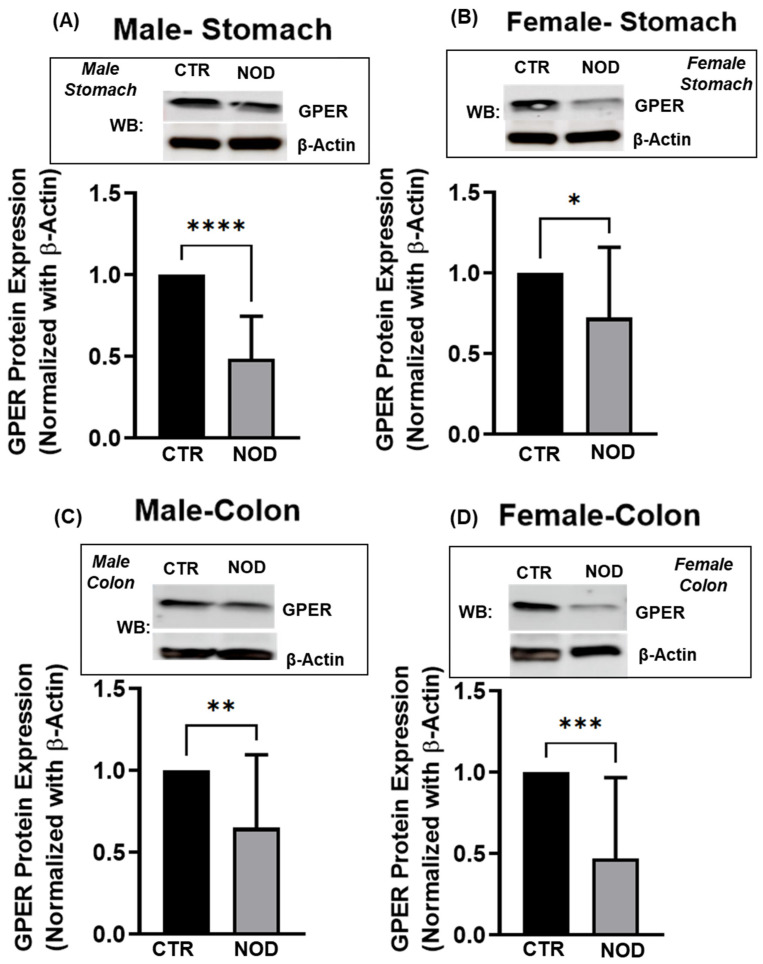
**GPER protein expression in CTR versus NOD male and female gastric and colonic smooth muscle tissues**. Total lysates were isolated from gastric and colonic smooth muscle tissue strips and subjected to Western blot analysis. (**A**–**D**) In gastric and colonic smooth muscle tissues, both male and female CTR mice exhibited significantly elevated GPER protein expression levels compared to those of NOD mice. The Western blot inserts represent one blot of one cohort of 4 biologically independent animals. The experiments were performed in triplicates, from samples collected from 20 biologically independent animals (5 male CTR, 5 female CTR, 5 male NOD, 5 female NOD). Results were deemed significant when *p* < 0.05; *t*-test analysis *p*-value (* *p* = 0.0314; ** *p* = 0.0087; *** *p* = 0.0008; **** *p* < 0.0001).

**Figure 6 ijms-25-05260-f006:**
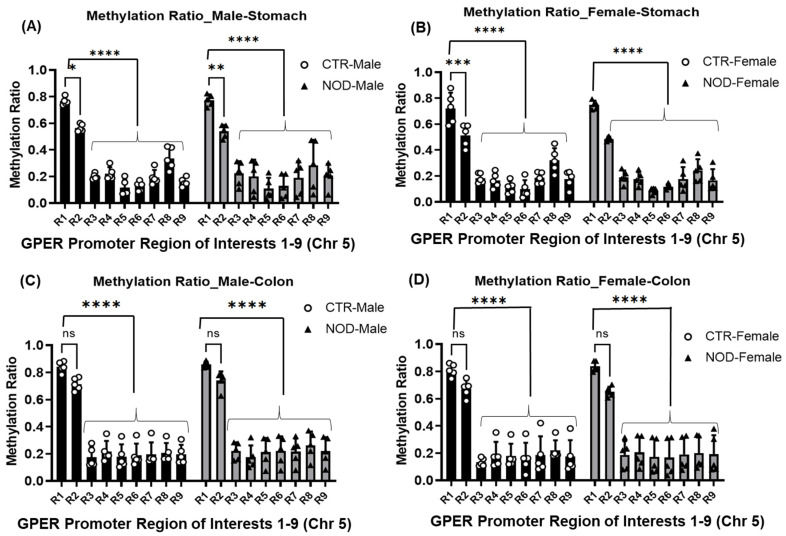
**Methylation ratio at regions R1-R9 in mouse CTR versus NOD male, female gastric and colonic smooth muscle tissues**. (**A**,**B**) In male and female gastric smooth muscle tissues, R1 exhibited a higher methylation ratio than R2–R9 in both CTR and NOD tissues. (**C**,**D**) In male and female colonic smooth muscle tissues, R1 exhibited a higher methylation ratio than R3–R9 in both CTR and NOD tissues. Very significant differences were found between R1 and R2–R9 in gastric smooth muscle and R3–R9 in colon smooth muscle. Results were deemed significant when *p* < 0.05. Two-way ANOVA analysis *p*-value (ns *p* > 0.05; * *p* = 0.0286; ** *p* = 0.0034; *** *p* = 0.0007; **** *p* < 0.0001) *n* = 5.

**Figure 7 ijms-25-05260-f007:**
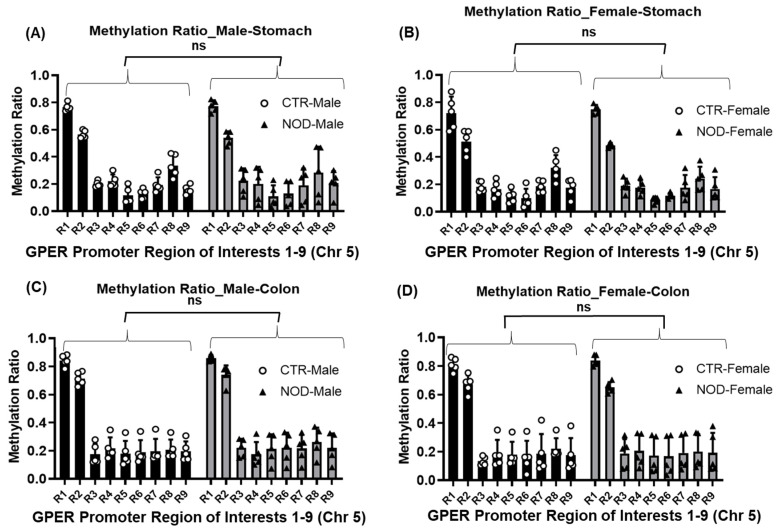
**Methylation ratio at regions R1-R9 in mouse CTR versus NOD male and female gastric and colonic smooth muscle tissues**. (**A**–**D**) No significant differences were found between homologous regions R1–R9 between CTR and NOD male and female mice in gastric and colonic smooth muscle. Results were deemed significant when *p* < 0.05. Two-way ANOVA analysis *p*-value (ns *p* > 0.05) *n* = 5.

**Figure 8 ijms-25-05260-f008:**
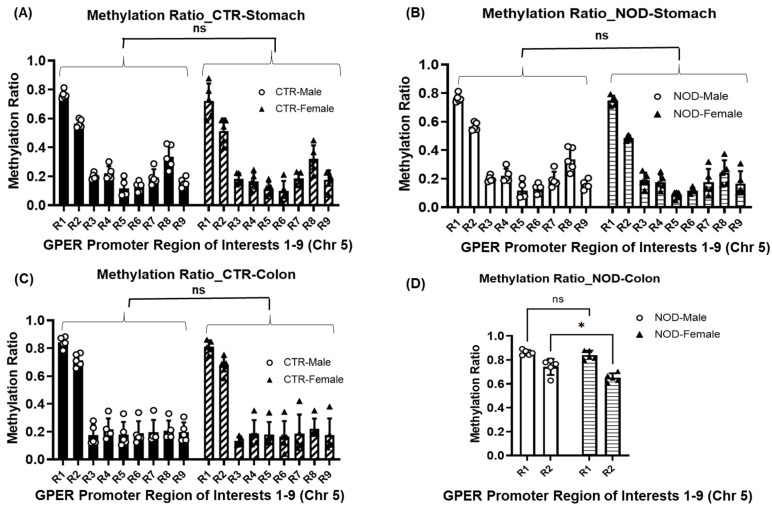
**Methylation ratios in gastric and colonic smooth muscle tissues of CTR and NOD mice between males and females**. (**A**,**B**) No significant differences were found between homologous regions R1–R9 between male and female CTR and NOD mice in gastric smooth muscle. (**C**) In colonic smooth muscle, there were no significant differences found between homologous regions R1–R9 between male and female CTR mice. (**D**) A significant difference was found between homologous region R2 between male and female NOD mice. Very significant differences were found between R1 and R3–R9. Results were deemed significant when *p* < 0.05. Two-way ANOVA analysis *p*-value (ns *p* > 0.05; * *p* < 0.0237) *n* = 5.

**Figure 9 ijms-25-05260-f009:**
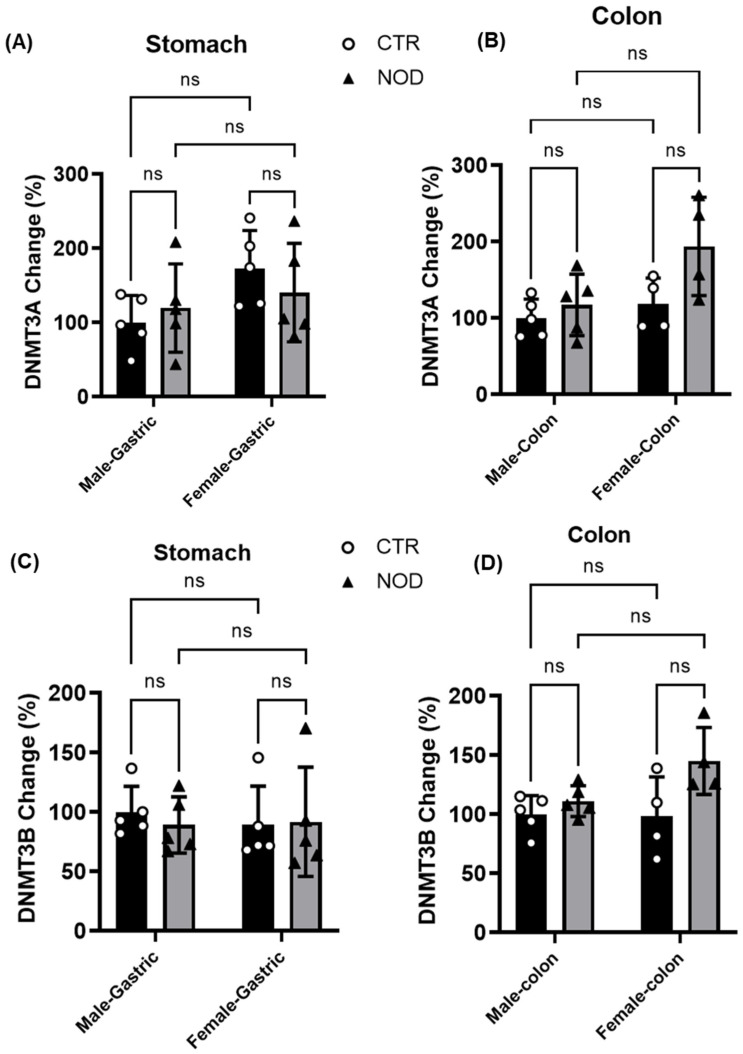
**DNMT levels in gastric and colonic smooth muscles of male and female CTR and NOD mice.** (**A**,**B**): DNMT3A levels in gastric and colonic smooth muscle in male and female mice were similar among all groups. Results were deemed significant when *p* < 0.05. Two-way ANOVA analysis *p*-value (ns *p* > 0.05; *n* = 4, 5). (**C**,**D**): DNMT3B levels in gastric and colonic smooth muscle in male and female mice were similar among all groups. Results were deemed significant when *p* < 0.05. Two-way ANOVA analysis *p*-value (ns *p* > 0.05; *n* = 4, 5).

**Figure 10 ijms-25-05260-f010:**
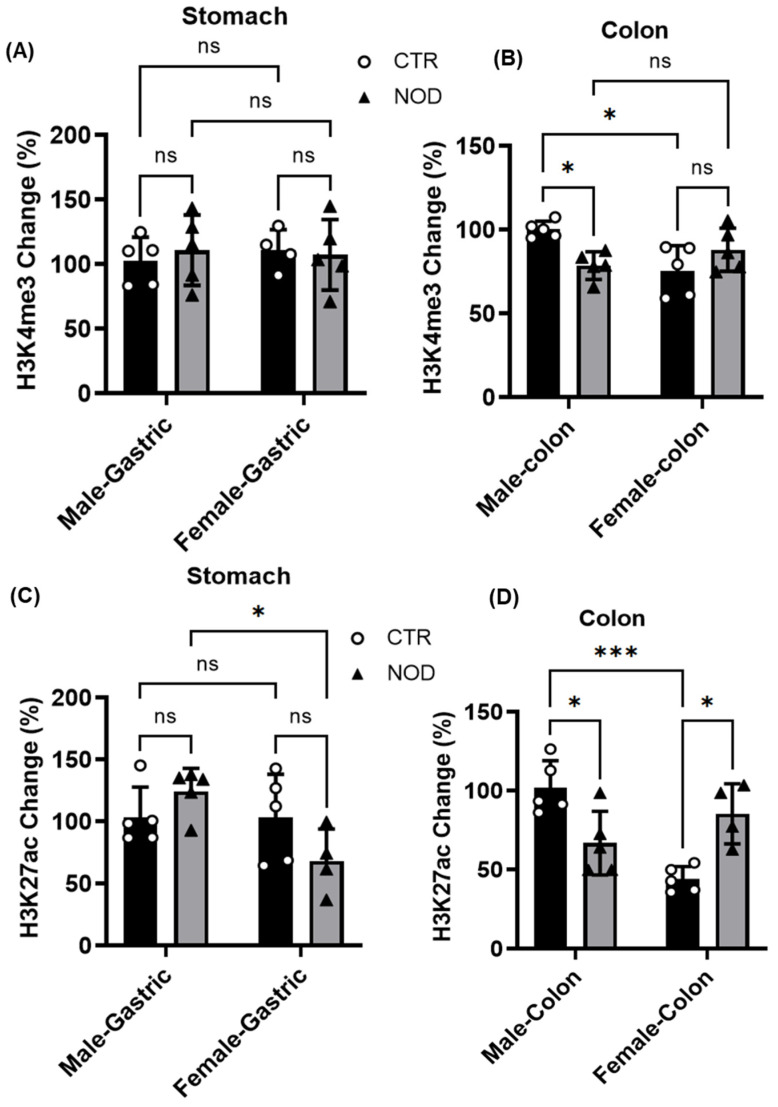
**H3K4me3 and H3K27ac levels in gastric and colonic smooth muscle of male and female CTR and NOD mice.** (**A**) H3K4me3 marks in the gastric smooth muscle of male and female CTR and NOD mice are similar. (**B**) In the colonic smooth muscle tissue, male CTR mice displayed higher levels of H3K4me3 than their NOD counterparts and female CTR mice. The results were considered statistically significant at *p* < 0.05. Two-way ANOVA analysis *p*-value (ns *p* > 0.05; * *p* = 0.0376; 0.015 CTR male vs. NOD male, CTR male vs. CTR female colon) *n* = 5. (**C**) H3K27ac levels in gastric smooth muscle tissue were similar between CTR and NOD male mice. However, female CTR mice exhibited significantly higher levels than female NOD mice, and H3K27ac levels in male CTR mice were lower than those in female CTR mice. Conversely, NOD males had higher H3K27ac levels than NOD females. (**D**) In the colon, NOD male mice had lower H3K27ac levels than CTR male mice. Female NOD mice displayed higher H3K27ac levels than female CTR mice. Female CTR mice exhibited lower H3K27ac levels than male CTR mice. The results were considered statistically significant at *p* < 0.05. Two-way ANOVA analysis *p*-value (ns *p* > 0.05; * *p* = 0.0197; 0.010 CTR male vs. NOD male, CTR female vs. NOD female colon; *** *p* = 0.0003) *n* = 5.

**Figure 11 ijms-25-05260-f011:**
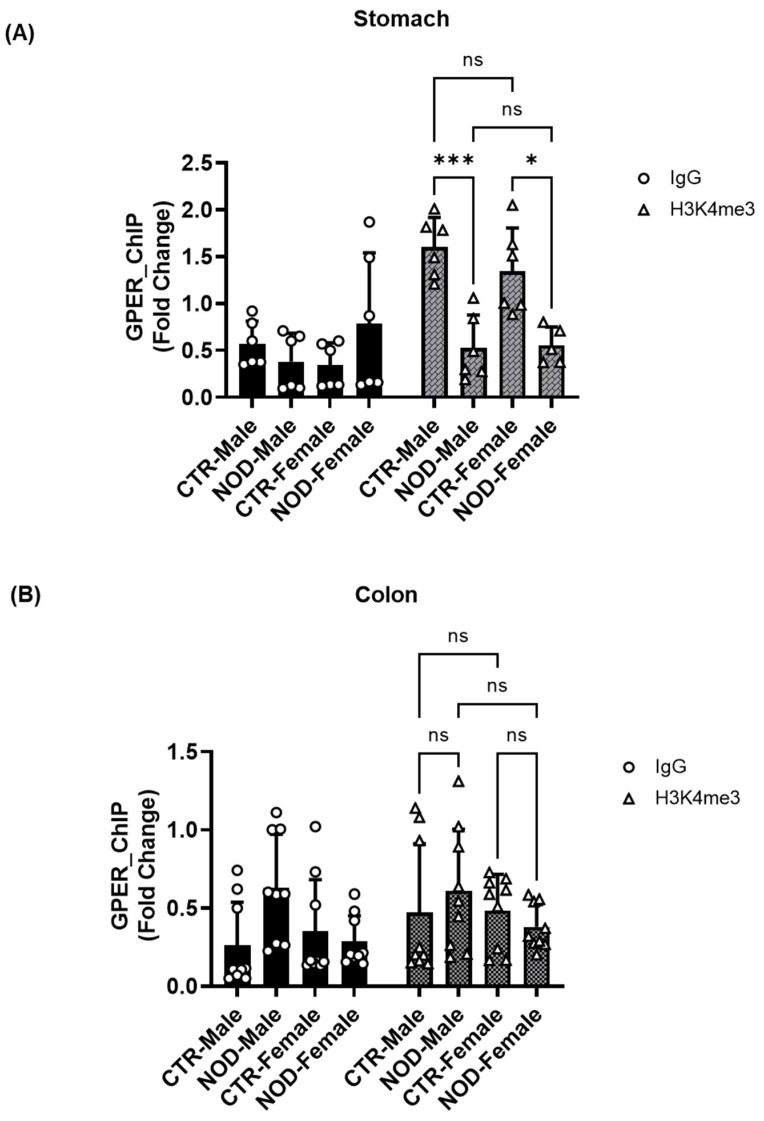
**H3K4me3 chromatin modifications at the GPER promoter region in CTR and NOD gastric and colonic smooth muscle.** (**A**) There was a significant reduction in the enrichment of H3K4me3 in the gastric tissues of both male and female NOD mice. Male CTR mice exhibited higher levels of H3K4me3 enrichment than female NOD mice. Results were deemed significant when *p* < 0.05. Two-way ANOVA analysis *p*-value (ns *p* > 0.05; * *p* = 0.042, CTR female vs. NOD female; *** *p* = 0.0008, CTR male vs. NOD male) *n* = 6 (**B**) Enrichment of H3K4me3 was similar in male and female NOD mice compared with male and female CTR mice. Results were deemed significant when *p* < 0.05. Two-way ANOVA analysis *p*-value (ns *p* > 0.05) *n* = 8.

**Figure 12 ijms-25-05260-f012:**
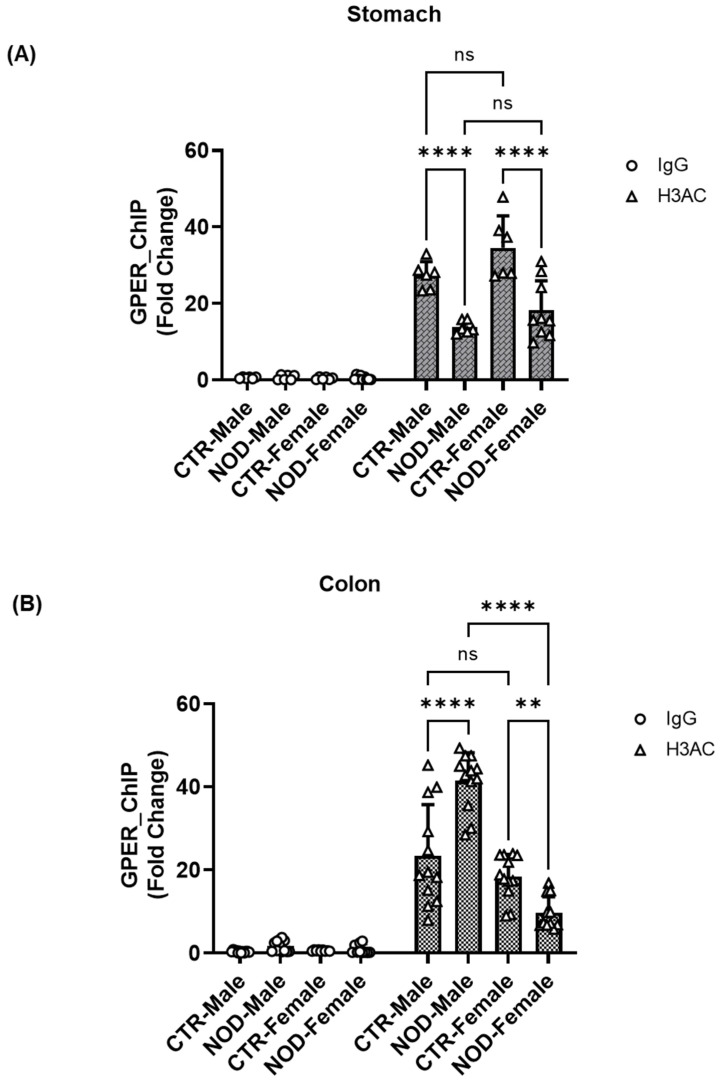
**H3ac chromatin modifications at the GPER promoter region in CTR and NOD gastric and colonic smooth muscle.** (**A**) In gastric smooth muscle, there was a reduction in the enrichment of the global histone acetylation marker (H3ac) in both male and female NOD mice relative to that in the CTR group. Male CTR mice showed higher H3ac enrichment than male NOD mice, whereas female CTR mice showed greater enrichment than female NOD mice. Results were deemed significant when *p* < 0.05. Two-way ANOVA analysis *p*-value (**** *p* < 0.00001, CTR male vs. NOD male; CTR female vs. NOD female); *n* = 6–9. (**B**) There was a significant increase in the enrichment of the H3ac in male NOD compared to male CTR and an increase in female CTR H3ac compared to female NOD. Higher enrichment of H3ac was also observed in male NOD mice than in female NOD mice. Results were deemed significant when *p* < 0.05. Two-way ANOVA analysis *p*-value (**** *p* < 0.0001, Control male vs. NOD male; NOD male vs. NOD female; ** *p* = 0.0076, CTR female vs. NOD female) *n* = 9–12.

**Figure 13 ijms-25-05260-f013:**
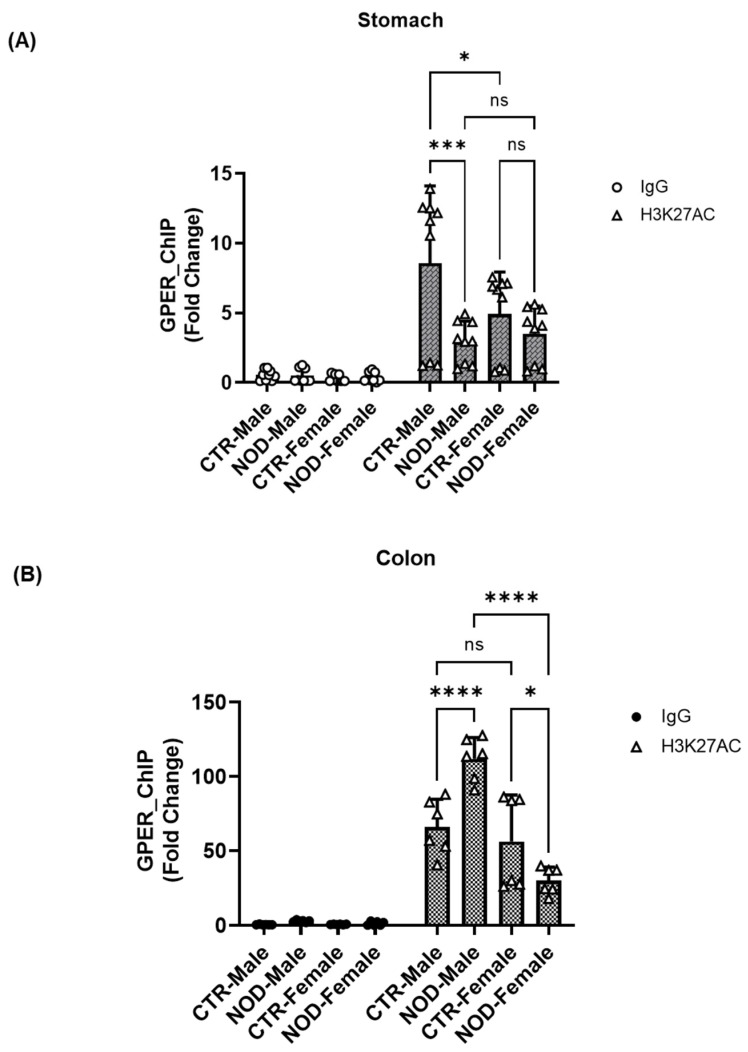
**H3K27ac chromatin modifications at the GPER promoter region in CTR and NOD gastric and colonic smooth muscle.** (**A**) Male NOD mice exhibited a reduction in the levels of the H3K27ac mark compared to male CTR mice in gastric smooth muscle, whereas female NOD mice had no significant H3K27ac levels compared to CTR female mice. Results were deemed significant when *p* < 0.05. Two-way ANOVA analysis *p*-value 2-way ANOVA analysis *p*-value (* *p* = 0.0393 CTR male vs. CTR female, *** *p* = 0.0001; CTR male vs. NOD male) *n* = 9. (**B**) H3K27ac enrichment followed a similar pattern. Compared with the male CTR, there was a significant increase in the enrichment of H3K27ac in male NOD colon tissue. In addition, there was a significant reduction in the enrichment of H3K27ac in NOD females compared to that in CTR females and NOD males. Results were deemed significant when *p* < 0.05. Two-way ANOVA analysis *p*-value (* *p* = 0.0483 CTR female vs. NOD female; **** *p* < 0.00001 CTR male vs. NOD male; NOD males vs. NOD females).

**Table 1 ijms-25-05260-t001:** Methylation ratio in the GPER promoter regions from region 1 (R1) to region 9 (R9) in CTR and NOD male and female gastric and colonic smooth muscle: (A) R1 vs. R2–R9 in CTR male and female gastric smooth muscle; (B) R1 vs. R2–R9 in NOD male and female gastric smooth muscle; (C) R1 vs. R2–R9 in CTR male and female colonic smooth muscle; (D) R1 vs. R2–R9 in NOD male and female colonic smooth muscle.

**(A)**					**(B)**				
**GPER Promoter Region Comparisons**	** *CTR-Male–Stomach* **		** *CTR-Female–Stomach* **		**GPER Promoter Region Comparisons**	** *NOD-Male–Stomach* **		** *NOD-Female–Stomach* **	
**R1 vs. R2–R9**	**Average**	**SEM**	**Average**	**SEM**	**R1 vs. R2–R9**	**Average**	**SEM**	**Average**	**SEM**
R1 vs. R2	−26.04	1.23 ^a^	−28.63	3.10 ^c^	R1 vs. R2	−30.31	1.50 ^b^	−34.93	0.74 ^d^
R1 vs. R3	−73.90	1.38 ^d^	−74.23	3.11 ^d^	R1 vs. R3	−71.60	4.78 ^d^	−74.66	2.83 ^d^
R1 vs. R4	−71.11	2.28 ^d^	−77.04	3.20 ^d^	R1 vs. R4	−74.57	6.93 ^d^	−76.38	2.94 ^d^
R1 vs. R5	−84.83	3.31 ^d^	−84.57	2.24 ^d^	R1 vs. R5	−86.12	4.66 ^d^	−88.44	1.41 ^d^
R1 vs. R6	−83.46	1.91 ^d^	−86.65	4.11 ^d^	R1 vs. R6	−83.67	4.95 ^d^	−84.29	1.25 ^d^
R1 vs. R7	−74.89	2.75 ^d^	−73.41	4.08 ^d^	R1 vs. R7	−76.03	6.86 ^d^	−76.73	5.27 ^d^
R1 vs. R8	−56.12	4.38 ^d^	−55.50	5.12 ^d^	R1 vs. R8	−63.85	10.54 ^d^	−68.00	4.74 ^d^
R1 vs. R9	−79.36	1.67 ^d^	−76.22	3.22 ^d^	R1 vs. R9	−73.60	4.90 ^d^	−77.91	4.76 ^d^
**(C)**					**(D)**				
**GPER Promoter Region Comparisons**	** *CTR-Male–Colon* **		** *CTR-Female–Colon* **		**GPER Promoter Region Comparisons**	** *NOD-Male–Colon* **		** *NOD-Female–Colon* **	
R1 vs. R2	−15.13	1.60 ^e^	−16.99	1.97 ^e^	R1 vs. R2	−13.70	3.48 ^e^	−22.24	1.98 ^e^
R1 vs. R3	−79.34	3.52 ^d^	−83.65	1.36 ^d^	R1 vs. R3	−74.21	3.46 ^d^	−77.87	5.32 ^d^
R1 vs. R4	−74.14	3.90 ^d^	−77.29	4.91 ^d^	R1 vs. R4	−79.67	4.46 ^d^	−75.66	5.97 ^d^
R1 vs. R5	−78.64	4.66 ^d^	−78.18	4.59 ^d^	R1 vs. R5	−74.85	5.41 ^d^	−79.52	6.72 ^d^
R1 vs. R6	−77.48	4.22 ^d^	−79.90	5.76 ^d^	R1 vs. R6	−74.04	5.73 ^d^	−80.29	7.10 ^d^
R1 vs. R7	−76.84	4.43 ^d^	−77.48	7.23 ^d^	R1 vs. R7	−74.59	5.22 ^d^	−77.52	6.40 ^d^
R1 vs. R8	−75.28	3.62 ^d^	−72.86	3.67 ^d^	R1 vs. R8	−69.23	5.45 ^d^	−76.30	6.30 ^d^
R1 vs. R9	−76.68	3.58 ^d^	−78.70	6.28 ^d^	R1 vs. R9	−74.09	5.42 ^d^	−77.32	7.21 ^d^

Region ^a^ = *p* < 0.01 (*); ^b^ = *p* < 0.001 (**); ^c^ = *p* < 0.0001 (***); ^d^ = *p* < 0.00001 (****); ^e^ = *p* > 0.05 (ns). R1 vs. R2–R9.

**Table 2 ijms-25-05260-t002:** Methylation ratio in the GPER promoter regions from region 1 (R1) to region 9 (R9) in CTR vs. NOD male and female gastric and colonic smooth muscle: (A) CTR male vs. NOD male in gastric smooth muscle, CTR female vs. NOD female in gastric smooth muscle; (B) CTR male vs. NOD male in colonic smooth muscle, CTR female vs. NOD female in colonic smooth muscle.

(A)					(B)				
CTR vs. NOD_Methylation Ratio (% Increase) ^e^ = ns *p* > 0.05					CTR vs. NOD_Methylation Ratio (% Increase) ^e^ = ns *p* > 0.05				
	*Male–Stomach*		*Female–Stomach*			*Male–Colon*		*Female–Colon*	
GPER Promoter Region Comparisons	Average	SEM	Average	SEM	GPER Promoter Region Comparisons	Average	SEM	Average	SEM
R1 vs. R1	1.07	1.64 ^e^	5.98	8.62 ^e^	R1 vs. R1	2.21	2.34 ^e^	3.68	2.90 ^e^
R2 vs. R2	−4.58	3.67 ^e^	−3.25	7.41 ^e^	R2 vs. R2	3.82	3.75 ^e^	−2.24	6.44 ^e^
R3 vs. R3	13.73	21.83 ^e^	10.10	21.66 ^e^	R3 vs. R3	41.78	28.73 ^e^	48.27	39.97 ^e^
R4 vs. R4	−10.26	26.36 ^e^	19.66	27.95 ^e^	R4 vs. R4	−16.35	18.11 ^e^	38.07	41.96 ^e^
R5 vs. R5	33.34	59.13 ^e^	−6.33	24.90 ^e^	R5 vs. R5	45.81	46.16 ^e^	22.60	47.08 ^e^
R6 vs. R6	10.69	40.98 ^e^	73.36	49.56 ^e^	R6 vs. R6	38.66	43.20 ^e^	26.05	44.17 ^e^
R7 vs. R7	−5.71	27.21 ^e^	−2.64	21.89 ^e^	R7 vs. R7	25.22	31.88 ^e^	58.19	58.08 ^e^
R8 vs. R8	−11.46	31.96 ^e^	−18.76	16.36 ^e^	R8 vs. R8	36.10	30.44 ^e^	2.79	32.44 ^e^
R9 vs. R9	38.82	34.93 ^e^	23.84	45.37 ^e^	R9 vs. R9	24.45	34.78 ^e^	62.73	64.41 ^e^

**Table 3 ijms-25-05260-t003:** Methylation ratio in the GPER promoter regions from region 1 (R1) to region 9 (R9) in male and female gastric and colonic smooth muscle: (A) male vs. female of gastric smooth muscle of CTR and NOD mice and (B) male vs. female of colonic smooth muscle of CTR and NOD mice.

(A)					(B)				
Male vs. Female_Methylation Ratio (% Increase) ^e^ = ns *p* > 0.05					Male vs. Female_Methylation Ratio (% Increase) ns *p* > 0.05; ^a^ = *p*< 0.01 (*)				
*CTR–Stomach*			*NOD–Stomach*		*CTR–Colon*			*NOD–Colon*	
GPER Promoter Region Comparisons	Average	SEM	Average	SEM	GPER Promoter Region Comparisons	Average	SEM	Average	SEM
R1 vs. R1	−5.84	5.92 ^e^	−2.94	4.40 ^e^	R1 vs. R1	−3.75	1.45 ^e^	−2.42	1.45 ^e^
R2 vs. R2	−8.74	7.96 ^e^	−9.13	5.13 ^e^	R2 vs. R2	−5.65	3.68 ^e^	−11.59	3.89 ^a^
R3 vs. R3	−7.71	9.14 ^e^	9.05	33.94 ^e^	R3 vs. R3	−14.34	15.70 ^e^	−10.54	22.92 ^e^
R4 vs. R4	−23.28	14.35 ^e^	63.07	68.33 ^e^	R4 vs. R4	−0.99	35.64 ^e^	35.46	48.07 ^e^
R5 vs. R5	16.50	37.29 ^e^	40.56	54.65 ^e^	R5 vs. R5	27.94	50.70 ^e^	27.45	75.38 ^e^
R6 vs. R6	−25.37	19.98 ^e^	51.32	55.56 ^e^	R6 vs. R6	5.75	45.51 ^e^	23.75	74.10 ^e^
R7 vs. R7	4.46	20.57 ^e^	82.04	83.99 ^e^	R7 vs. R7	2.78	39.78 ^e^	36.15	72.40 ^e^
R8 vs. R8	−3.48	10.85 ^e^	72.65	79.70 ^e^	R8 vs. R8	16.86	26.79 ^e^	1.87	47.34 ^e^
R9 vs. R9	9.25	16.15 ^e^	−4.45	24.92 ^e^	R9 vs. R9	2.84	43.04 ^e^	37.11	82.48 ^e^

## Data Availability

Data are available in the Gene Expression Omnibus (GEO) datasets. Targeted bisulfite sequencing dataset is available under accession number GSE255236.

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
