# Peer review of "Epigenetic Modulation of GPER Expression in Gastric and Colonic Smooth Muscle of Male and Female Non-Obese Diabetic (NOD) Mice: Insights into H3K4me3 and H3K27ac Modifications"

_ijms, 2024, doi:10.3390/ijms25105260_

Round 1
Reviewer 1 Report
Comments and Suggestions for Authors
This manuscript by Hixon and colleagues examines the expression of the G-Protein coupled estrogen receptor, Gper in the non-obese diabetes (NOD) mouse model. They find significant downregulation of Gper mRNA in NOD mice stomach and colon smooth muscle tissue. They investigate the potential epigenetic regulation of Gper by promoter CpG methylation and levels of histone modifications and find sex-specific epigenetic associations. In particular, fitting with the observed downregulation of Gper, reduced levels of ‘active’ histone modifications are observed at the Gper promoter in gastric muscle of the NOD mice. Surprisingly in the colon, male mice show increased level of active histone acetylation at the Gper promoter, while females show the expected reduction, correlating with the decreased expression of Gper.
This manuscript documents potentially interesting results concerning the regulation of Gper in the gastric and colon of diabetic mice. However, I have a number of concerns about the data analysis and presentation in the manuscript detailed below, which must be addressed before publication. In addition, the main conclusions of the manuscript are not clear, largely due to the inclusion of insignificant and irrelevant analysis as detailed below. The authors should streamline the text to focus on the main findings.
1. For the analysis of DNA Methylation ratios the authors make inappropriate comparisons. Frequently they compare conditions with more than one variable – for example they compare Region 1 in Control mice to Region 2 in NOD mice. This analysis is meaningless and should be removed from the manuscript. A similar comparison is made in Figure 11, where female NOD mice are compared to male Controls.
2. While a limited analysis of Gper promoter CpG Counts is warranted as a control and to serve as a basis for the selection of regions R1 and R2, the authors go too far in the analysis of potential differences in CpG counts between conditions. Panels detailing total CpG counts in various conditions are found in Figures 3 – 10, which is far too much and diverts from the main results of the manuscript. For example, they state ‘Remarkably, no statistically significant difference was found between R1 and R2 in the in (sic) both male and female colonic smooth muscle from the NOD and CTR groups (page 12, line 395)’, referring to Figures 6A and B, which show Total CpG counts. Why is this remarkable? Why would the authors expect the genotypes of the Control and NOD mice to differ, particularly in the number of CpGs in the promoter region of the Gper gene? The CpG counts analysis should be removed from the manuscript, except for the rationale behind selection of R1 and R2.
3. For the qPCR of Gper mRNA (Figures 1+2), the authors present the data with all replicates in the male (panels A+B) or control (panels C+D) normalised to 1. If the statistics are calculated based on these values they are inaccurate as it will not take into account the full variance in the dataset. Fold changes should be calculated in all samples (including male and control) to the average of the male or control condition, if normalisation to the male or control is to be performed. In this way, the variance in the data is preserved. In addition, they describe the up or downregulation expressed as a percentage with errors in the text (e.g. 297.1 ± 104.0% higher in NOD females). What does this error refer to, as it doesn’t fit with the error bars presented in the Figures?
4. The western blots in Figures 1 and 2 are not described in the legends. Are these blots representative of how many independent experiments?
5. The authors state (page 23, line 567): ‘Specifically, we assessed the levels of H3K4me3 and H3K27ac, which are known to significantly affect gene expression.’ This is not true; it is currently unclear whether these modifications are instructive for gene expression or correlated with gene expression. Please change the text. The statement (line 743) ‘Establishing the H3K4me3 mark requires writing the H3K27ac mark first’ is also not correct; it has not been shown the H3K27ac is required for H3K4me3, only that H3K27ac establishment can lead to the establishment of H3K4me3, which are very different.
6. In the conclusion, the sentence ‘Enrichment of the GPER promoter region also helps explain these differences’ is unclear. Enrichment of what in the promoter region?
7. The phrase ‘our findings are in line with the observations of Zhao et al., who reported that H3K4me3 and not H3K27ac is required for gene activation’ should be amended because Zhao et al did not report that H3K4me3 and not H3K27ac is required for gene activation. Firstly, they only study whether the installation of these marks can lead to gene activation, not whether they are required for gene activation (which would necessitate experimental removal of these marks) and secondly they actually conclude the opposite, that installation of K27ac can lead to gene activation, not H3K4me3.
8. On page 25, they refer to the ‘more specific' histone trimethylation mark (H3K4me3). What do they mean by 'more specific'?
Comments on the Quality of English LanguageMinor editing needed
Author Response
We deeply appreciate the invaluable feedback and insightful recommendations provided by the reviewers. Thank you sincerely for your thorough review.
In accordance with the reviewers' comments, we have meticulously attended to each concern in the revised manuscript. All modifications have been diligently tracked and highlighted in red for your convenience. In response to the reviewer's document, reviewer’s comments are presented in plain text, while our corresponding replies are denoted by italicization.
#REVIEWER 1
Comments and Suggestions for Authors
This manuscript by Hixon and colleagues examines the expression of the G-Protein coupled estrogen receptor, Gper in the non-obese diabetes (NOD) mouse model. They find significant downregulation of Gper mRNA in NOD mice stomach and colon smooth muscle tissue. They investigate the potential epigenetic regulation of Gper by promoter CpG methylation and levels of histone modifications and find sex-specific epigenetic associations. In particular, fitting with the observed downregulation of Gper, reduced levels of ‘active’ histone modifications are observed at the Gper promoter in gastric muscle of the NOD mice. Surprisingly in the colon, male mice show increased level of active histone acetylation at the Gper promoter, while females show the expected reduction, correlating with the decreased expression of Gper.
This manuscript documents potentially interesting results concerning the regulation of Gper in the gastric and colon of diabetic mice. However, I have a number of concerns about the data analysis and presentation in the manuscript detailed below, which must be addressed before publication. In addition, the main conclusions of the manuscript are not clear, largely due to the inclusion of insignificant and irrelevant analysis as detailed below. The authors should streamline the text to focus on the main findings.
- For the analysis of DNA Methylation ratios the authors make inappropriate comparisons. Frequently they compare conditions with more than one variable – for example, they compare Region 1 in Control mice to Region 2 in NOD mice. This analysis is meaningless and should be removed from the manuscript. A similar comparison is made in Figure 11, where female NOD mice are compared to male Controls.
Response: We acknowledge the reviewer's insightful critique regarding certain unnecessary comparisons in our manuscript. Accordingly, we have diligently revised the text, eliminating all superfluous comparisons. Our focus now remains on comparisons solely between control and NOD mice within the same sex, as well as between male and female mice within the NOD or control groups, as depicted in Figures 1-15 and supplementary figures. We express gratitude to the reviewer for bringing this issue to our attention, thereby enhancing the clarity and relevance of our work.
- While a limited analysis of Gper promoter CpG Counts is warranted as a control and to serve as a basis for the selection of regions R1 and R2, the authors go too far in the analysis of potential differences in CpG counts between conditions. Panels detailing total CpG counts in various conditions are found in Figures 3 – 10, which is far too much and diverts from the main results of the manuscript. For example, they state ‘Remarkably, no statistically significant difference was found between R1 and R2 in the in (sic) both male and female colonic smooth muscle from the NOD and CTR groups (page 12, line 395)’, referring to Figures 6A and B, which show Total CpG counts. Why is this remarkable? Why would the authors expect the genotypes of the Control and NOD mice to differ, particularly in the number of CpGs in the promoter region of the Gper gene? The CpG counts analysis should be removed from the manuscript, except for the rationale behind selection of R1 and R2.
Response: We appreciate the reviewer's astute observation and agree that the detailed statistical analysis of total CpG counts was unnecessary. Accordingly, we have omitted these comparisons, prioritizing the methylation ratio, the central focus of our study, and provided data spanning Regions 1 to 9 for thoroughness.
- For the qPCR of Gper mRNA (Figures 1+2), the authors present the data with all replicates in the male (panels A+B) or control (panels C+D) normalised to 1. If the statistics are calculated based on these values they are inaccurate as it will not take into account the full variance in the dataset. Fold changes should be calculated in all samples (including male and control) to the average of the male or control condition, if normalisation to the male or control is to be performed. In this way, the variance in the data is preserved. In addition, they describe the up or downregulation expressed as a percentage with errors in the text (e.g. 297.1 ± 104.0% higher in NOD females). What does this error refer to, as it doesn’t fit with the error bars presented in the Figures?
Response: We have refined the results section and supplemented it with a comprehensive method, referencing BioRad's data analysis protocol for clarity and accuracy .(line 163-195 and line 513-530)
- The western blots in Figures 1 and 2 are not described in the legends. Are these blots representative of how many independent experiments?34ws
Response: The description of the western blot results has been included in the figure legends of figures 1 and 2. The blots represent diagrams from 1 biologically-independent experiment. All the biologically-independent experiments blots were provided.
- The authors state (page 23, line 567): ‘Specifically, we assessed the levels of H3K4me3 and H3K27ac, which are known to significantly affect gene expression.’ This is not true; it is currently unclear whether these modifications are instructive for gene expression or correlated with gene expression. Please change the text.
Response: We acknowledge the reviewer's observation regarding the statement's prematurely conclusive tone concerning ongoing investigations. Consequently, we have revised it to a more fitting form, as detailed in the manuscript.
The statement (line 743) ‘Establishing the H3K4me3 mark requires writing the H3K27ac mark first’ is also not correct; it has not been shown the H3K27ac is required for H3K4me3, only that H3K27ac establishment can lead to the establishment of H3K4me3, which are very different.
Response: We agree with the reviewer’s argument and have paraphrased the statement to a more appropriate form.
- In the conclusion, the sentence ‘Enrichment of the GPER promoter region also helps explain these differences’ is unclear. Enrichment of what in the promoter region?
Response: The sentence has been paraphrased and more information was added to improve the comprehension of its meaning.
- The phrase ‘our findings are in line with the observations of Zhao et al., who reported that H3K4me3 and not H3K27ac is required for gene activation’ should be amended because Zhao et al did not report that H3K4me3 and not H3K27ac is required for gene activation. Firstly, they only study whether the installation of these marks canlead to gene activation, not whether they are required for gene activation (which would necessitate experimental removal of these marks) and secondly they actually conclude the opposite, that installation of K27ac can lead to gene activation, not H3K4me3.
Response: We appreciate the reviewer for this important observation. We believe that the previous statement was a mistake and has been corrected accordingly.
- On page 25, they refer to the ‘more specific' histone trimethylation mark (H3K4me3). What do they mean by 'more specific'?
Response: Using the term more specific was inappropriate and has been eliminated accordingly.
Reviewer 2 Report
Comments and Suggestions for Authors
This is a study to investigate if one of the complications which is more prevalent in females with type I diabetes (reduced gastric motility) may be related to the changes in G-protein coupled estrogen receptor (GPER) expression.
It is an ex-vivo study on two animal models including type I non-obese diabetic NOD mice (ShiLtJ) and as the control (CTRL), NOD B10Sn-H2b/J mice that do not develop diabetes.
Aspects examined, in gastric and colonic smooth muscle cell (SMS) tissues, include GPER mRNA and protein expression (the latter by WB or Western blots), and genetic and epigenetic marks which can influence mRNA and consequently GPER protein expression.
Respectfully, while there are some findings of statistical significance, there are very few substantive findings that make for a useful contribution to this area of research and there are too many issues with experimental design and interpretations to warrant publication, at least in in my view. The conclusion itself gives an idea of the very limited scope of the findings. In terms of scientific writing, there is much to be improved-there is a lot of padding- i.e. repetition, and a lack of clarity and direction. A handful of these significant shortfalls are listed below. But there are other significant issues which would take much more time to address, and frankly the paper is difficult enough to follow. From the point of view of presentation alone, it is clear that a thorough and meticulous re-write is in order.
Results
GPER mRNA and protein expression (Fig 1 and 2)
Figure 1 and 2 show decreased GPER mRNA expression in NOD gastric and colonic SMC tissue. There are several issues here. Figure labelling and the legend is poor in regard to the Western blot results (WB). Relating to the methodology, there is only one control gene for the mRNA studies which is no longer considered sufficient, as has been evident now for several decades-one needs to have several mRNA control genes, and while the authors purport to show the mRNA is also associated with decreased GPER protein expression as evident from the one set of WB results shown, there is no statistical evidence to show that the WB results are statistically significant. If that data was available, then blots from all experiments should have also been included in the supplementary data.
A minor point, but concerning the comparison between female and male control stomachs: Figure 1A….the text on page 7 cites figures of 23% (female) and 49% (male); i.e. 23/49x100=46% relative to males in the females, and yet the graph in Figure 1 A shows females are just above 50% of male values. This needed an explanation.
At the same time, while the text says there was no difference between control males and females, the spread of data in the females is such that one cannot say with any certainty whether a difference exists-there just isn’t enough statistical power for that (and the same applies to Fig 2A). There is also a lack of transparency as to how the males and females are compared…at the very least the results of values relative to beta actin in males and females should be supplied as supplementary information for review. The use of relative values is opaque—it suggests some sort of matching between specific males and females. Likewise for the other figures….how this was done is not specified in the methods.
There are similar issues elsewhere; e.g. for Fig 1D, the text on page 7 should say as it does for 1C in males, 64.8% lower in NOD females
GPER promotor studies (CpG sites and methylation status-Figs 3 to 10)
Purported to have shown decreased mRNA expression (inadequately in my view given the lack of control mRNA genes), the authors go on to explore differences in the GPER promoter region, including any sequence differences (in CpG sites) and methylation patterns that might explain and change in mRNA expression.
Figure 3-Figure 4. Gastric SMC tissue
a)The authors state at the bottom of page 34 that they examined the CpG sites in the GPER promoter by sequencing of nine different regions (R1 to R9). Why is the location of those regions not specified. They certainly don’t correspond to the ~500 bp promoter sequence defined in the methods-presumably it’s an upstream region, but what is the location?
Much later in the discussion, the authors mention that their results has added much to the field on the composition of the GPER promoter-how can it when the location is not specified.
b) Two of the regions, (R1 and R2) have a higher CpG count than other regions- and that becomes a focus as a higher density might be more prone to epigenetic silencing via methylation. What strikes the authors as interesting is that were no differences either in R1 or R2 CpG density between any groups of mice, male female, NOD or control mice. But isn’t that precisely what one would expect from inbred strains of mice? It would have been worrisome indeed if there were sequence differences.
c) Concerning the methylation of R1 and R2, a consistent finding is a higher methylation ratio of R1 than R2 regions in both sexes of the CTRL and NOD mice. Having shown that why make inappropriate comparisons between methylation status of the R1 region of CTRL males with the methylation status of the R2 region of NOD females? These types of inappropriate comparisons are a recurring theme throughout the paper, not just this section.
The appropriate comparisons should firstly always be of the same regions with the appropriate controls with the authors fundamental questions in mind what explains the increased predisposition to GI dysmotility in female diabetics in particular. So the one would think that the relevant comparisons in terms of priority/thoroughness might be.
male NOD vs female NOD,
male CTRL vs female CTL,
female CTRL vs female NOD,
male CTRL vs male NOD for completion
Figures 5-6.Colonic SMS tissue
This section is almost identical to that for issues identified above but now for colonic SMC tissue and highlights an issue with how incredibly repetitious this paper can be. I wonder why there was a need to sequence the R1 to R9 regions in colonic tissue-one would not expect any difference whatsoever in colonic vs gastric DNA sequences, surely!
Figures 7 and on.
Finally, it’s not until we get to figure 7 that we see some semblance of appropriately prioritized/emphasized comparisons comparing male and female mice, within the CTRL and NOD mice. But we also again see this recurring pattern of inappropriate comparisons – and why, is this a search for statistical significance? I cite: “the methylation ratio of R1 in male gastric smooth muscle was higher than the R2 methylation ration of in female gastric smooth muscle”
Comments on the Quality of English LanguageI've noted: "Extensive editing of English language required" - this is principally because there is in my view so much repetition, and lack of direction.
Author Response
We deeply appreciate the invaluable feedback and insightful recommendations provided by the reviewers. Thank you sincerely for your thorough review.
In accordance with the reviewers' comments, we have meticulously attended to each concern in the revised manuscript. All modifications have been diligently tracked and highlighted in red for your convenience. In response to the reviewer's document, reviewer’s comments are presented in plain text, while our corresponding replies are denoted by italicization.
REVIEWER 2
Comments and Suggestions for Authors
This is a study to investigate if one of the complications which is more prevalent in females with type I diabetes (reduced gastric motility) may be related to the changes in G-protein coupled estrogen receptor (GPER) expression.
It is an ex-vivo study on two animal models including type I non-obese diabetic NOD mice (ShiLtJ) and as the control (CTRL), NOD B10Sn-H2b/J mice that do not develop diabetes.
Aspects examined, in gastric and colonic smooth muscle cell (SMS) tissues, include GPER mRNA and protein expression (the latter by WB or Western blots), and genetic and epigenetic marks which can influence mRNA and consequently GPER protein expression.
Respectfully, while there are some findings of statistical significance, there are very few substantive findings that make for a useful contribution to this area of research and there are too many issues with experimental design and interpretations to warrant publication, at least in my view. The conclusion itself gives an idea of the very limited scope of the findings. In terms of scientific writing, there is much to be improved-there is a lot of padding- i.e. repetition, and a lack of clarity and direction. A handful of these significant shortfalls are listed below. But there are other significant issues which would take much more time to address, and frankly the paper is difficult enough to follow. From the point of view of presentation alone, it is clear that a thorough and meticulous re-write is in order.
Results
GPER mRNA and protein expression (Fig 1 and 2)
Figure 1 and 2 show decreased GPER mRNA expression in NOD gastric and colonic SMC tissue. There are several issues here. Figure labelling and the legend is poor in regard to the Western blot results (WB). Relating to the methodology, there is only one control gene for the mRNA studies which is no longer considered sufficient, as has been evident now for several decades-one needs to have several mRNA control genes, and while the authors purport to show the mRNA is also associated with decreased GPER protein expression as evident from the one set of WB results shown, there is no statistical evidence to show that the WB results are statistically significant. If that data was available, then blots from all experiments should have also been included in the supplementary data.
Response: The figure labels and legends of figures 1 and 2 have been corrected accordingly lines 856-878). We have added to the supplementary data the results comparing GPER expression patterns normalized using two more control genes (18S rRNA and GAPDH) (Figure 1 and 2; supplementary figures 1 and 2). Additional WB results have also been included in the supplementary data (supplementary). Thanks to the reviewer.
A minor point, but concerning the comparison between female and male control stomachs: Figure 1A….the text on page 7 cites figures of 23% (female) and 49% (male); i.e. 23/49x100=46% relative to males in the females, and yet the graph in Figure 1 A shows females are just above 50% of male values. This needed an explanation.
Response: We modified the text.
At the same time, while the text says there was no difference between control males and females, the spread of data in the females is such that one cannot say with any certainty whether a difference exists-there just isn’t enough statistical power for that (and the same applies to Fig 2A).
Response: We agree that, based on the spread of the data, it is difficult to tell whether any significant differences exist. We have made the text and graphs clearer. Concerning this, our raw data is available for scrutiny.
There is also a lack of transparency as to how the males and females are compared…at the very least the results of values relative to beta actin in males and females should be supplied as supplementary information for review. The use of relative values is opaque—it suggests some sort of matching between specific males and females. Likewise for the other figures….how this was done is not specified in the methods.
Response: we added the values relative to beta actin and other endogenous controls. About the matching, they were randomly matched, since they were of the same age, arrived at the same time, from the same company, and kept in the same environmental condition.
There are similar issues elsewhere; e.g. for Fig 1D, the text on page 7 should say as it does for 1C in males, 64.8% lower in NOD females
Response: This correction has been noted and effected accordingly.
GPER promotor studies (CpG sites and methylation status-Figs 3 to 10)
Purported to have shown decreased mRNA expression (inadequately in my view given the lack of control mRNA genes), the authors go on to explore differences in the GPER promoter region, including any sequence differences (in CpG sites) and methylation patterns that might explain and change in mRNA expression.
Figure 3-Figure 4. Gastric SMC tissue
- a) The authors state at the bottom of page 34 that they examined the CpG sites in the GPER promoter by sequencing of nine different regions (R1 to R9). Why is the location of those regions not specified? They certainly don’t correspond to the ~500 bp promoter sequence defined in the methods-presumably it’s an upstream region, but what is the location?
Response: The regions R1 – R9 are specific locations in the GPER promoter that contain a CpG residue. All the 9 regions are, therefore, part of the ~ 500 bp GPER promoter and only contain one CpG site each. On the other hand, the total count refers to the number of times a CpG at a particular position is sequenced, which increases our confidence that it is indeed a CpG site in the GPER promoter. Given that regions 1 and 2 have higher counts than other regions, we became more confident that these two regions are indeed potential sites of methylation in the GPER promoter. Thus, we decided to compare the methylation ratios of R1 and R2. The methylation ratios were calculated by dividing the number of methylated cytosine(s) by the total number of cytosine(s) found in the reads. In short, we identified 9 CpG sites in the GPER promoter sequence among which we selected R1 and R2 that we think might have the highest potential for methylation. The locations of these 9 CpG sites have been included in the methodology section under the subsection “GPER promoter sequence”. Also we provided all the regions of methylation (lines 569-583).
Much later in the discussion, the authors mention that their results has added much to the field on the composition of the GPER promoter-how can it when the location is not specified.
Response: The locations have been specified.
- b) Two of the regions, (R1 and R2) have a higher CpG count than other regions- and that becomes a focus as a higher density might be more prone to epigenetic silencing via methylation. What strikes the authors as interesting is that were no differences either in R1 or R2 CpG density between any groups of mice, male female, NOD or control mice. But isn’t that precisely what one would expect from inbred strains of mice? It would have been worrisome indeed if there were sequence differences.
Response: We have removed any results comparing CpG counts from the manuscript and have instead focused on methylation ratio differences, which are part of the epigenetic mechanisms we studied. Thanks to the reviewer.
- c) Concerning the methylation of R1 and R2, a consistent finding is a higher methylation ratio of R1 than R2 regions in both sexes of the CTRL and NOD mice. Having shown that why make inappropriate comparisons between methylation status of the R1 region of CTRL males with the methylation status of the R2 region of NOD females? These types of inappropriate comparisons are a recurring theme throughout the paper, not just this section.
Response: We appreciate the reviewer for this important observation. We admit that such comparisons are inappropriate and we, therefore, removed them throughout the manuscript.
The appropriate comparisons should firstly always be of the same regions with the appropriate controls with the authors fundamental questions in mind what explains the increased predisposition to GI dysmotility in female diabetics in particular. So the one would think that the relevant comparisons in terms of priority/thoroughness might be.
male NOD vs female NOD,
male CTRL vs female CTL,
female CTRL vs female NOD,
male CTRL vs male NOD for completion
Response: We strongly commend the reviewer for this important suggestion. This will indeed make the work more presentable. We have, therefore, adopted the suggested comparisons throughout the manuscript and have changed the presentation of the results based on the suggested comparisons throughout the manuscript.
Figures 5-6.Colonic SMS tissue
This section is almost identical to that for issues identified above but now for colonic SMC tissue and highlights an issue with how incredibly repetitious this paper can be. I wonder why there was a need to sequence the R1 to R9 regions in colonic tissue-one would not expect any difference whatsoever in colonic vs gastric DNA sequences, surely!
Response: We admit that this was a mistake and we have also removed any comparisons involving CpG counts as we did for the gastric tissue. We appreciate the reviewer once again for his/her selfless efforts to make our work better.
Figures 7 and on.
Finally, it’s not until we get to figure 7 that we see some semblance of appropriately prioritized/emphasized comparisons comparing male and female mice, within the CTRL and NOD mice. But we also again see this recurring pattern of inappropriate comparisons – and why, is this a search for statistical significance? I cite: “the methylation ratio of R1 in male gastric smooth muscle was higher than the R2 methylation ration of in female gastric smooth muscle”
Response: Once again, we admit that such comparisons are inappropriate and hence have been eliminated throughout the results.
Comments on the Quality of English Language
I've noted: "Extensive editing of English language required" - this is principally because there is in my view so much repetition, and lack of direction.
Response: The English language has been improved throughout the manuscript as suggested by the reviewer.
Round 2
Reviewer 1 Report
Comments and Suggestions for Authors
Please view the attached file.

Minor editing of the English is required. For example, on page 13: 'when was used B-actin as a reference gene'.
Author Response
Manuscript, (ijms-2917089), Titled, Epigenetic Modulation of GPER Expression in Gastric and Colonic Smooth Muscle of Male and Female Non-obese Diabetic (NOD) Mice: Insights into H3K4me3 and H3K27ac Modifications.
.
Response to the reviewers:
We are profoundly grateful for the invaluable feedback and insightful recommendations generously provided by the reviewers. We extend our sincere thanks for the thorough review.
In response to the reviewers' comments, we have diligently addressed each concern in the revised manuscript. Every modification has been meticulously tracked and highlighted in red for your convenience. Reviewer comments are presented in plain text, with our corresponding replies denoted by italicization
Reviewer’s Comments:
The manuscript has significantly improved. Most of my comments have been addressed. However I still have a few concerns, as detailed below:
- Overall I find the text difficult to read. Often it reads like a list of disconnected results and the reader struggles to find the important results and the thread of the manuscript. While am I not suggesting to remove negative results, perhaps sometimes explanation of non-significant differences can be removed from the text to improve the readability (but remain in the figures).
Response: We acknowledge the reviewer's insightful critique and have refined the text to make it more readable.
- While I appreciate the authors removing the CpG counts information for the Gper promoter, there is now too much discussion of the methylation difference between R1 and R2-9 in the various conditions (Figure 4-10). As this is a consistent effect, this part of the text could easily be streamlined.
Response: We have summarized the methylation results section to make it easier to follow.
- Regarding the Western blot data, the authors mention in the response to the reviewers that the data shown in Figures 1 and 2 are representative of 1 biologically independent experiment. If this is true, this is not adequate reproducibility for publication and these experiments should be repeated. However, in the Supplementary information they show 4-5 blots for the same experiments in Schemes 4-8. It is still not described in the figure legend to Figures 1 and 2 or in the Methods section how many biological independent experiments the Western blotting data represent. Thus, this needs to be clarified and repeated if necessary.
Response: a) The Western blot experiments were performed in triplicates from samples collected from 20 biologically independent animals ( 5 male CTR, 5 female CTR, 5 male NOD, 5 female NOD). The Western blot inserts represent one blot of one cohort of biologically independent animals.
b) We have adjusted the Methods section to reflect on the number of biologically independent experiments and have modified the figure legends.
The quantification of the Western blotting data shown in Supplementary Scheme 9, shows no statistically significant difference between control and NOD mice, except in the male stomach. However, the authors describe in this legend that GPER protein expression was significantly lower in both male and female stomach and colon, considering p < 0.05. This is incorrect and must be addressed. In addition, the section in the main text should be amended to indicate that the protein is only significantly decreased in the male stomach and this should be discussed accordingly. Lastly, does the quantification include normalization to the control Beta-Actin? This is not indicated and, if not, normalization to the loading control should be performed to account for differences in total protein loading in the gels.
Response: We have made the necessary changes to the text to reflect the results represented. Quantification was normalized to Beta-Actin, after further review, we have updated the results.
- Unfortunately I am still not satisfied with the explanation the authors provide regarding the statistical analysis of the qPCR results. Are the statistics calculated based on the normalized fold change values (when all control samples = 1?). This is still not clear from the new methods. If so, this is incorrect as the variance is incorrectly reduced to 0 in the control. Taking into account the variability of a particular group of samples is essential for calculating statistics. To avoid this, one can normalize each individual control sample to the average of the control samples, in order to preserve the variance in the data.
Response: The smooth muscle samples isolated from CTR and NOD mice sacrificed on the same day were processed together as matched pairs (CTR male vs NOD male, CTR female vs NOD female) to ensure identical ages for comparison (Explained in the methods also). Recently our lab published similar type of calculations in Nature/Scientific Report Journal. Also, many of the manuscripts working with animal, plant and human samples also used the same method. Please see the following manuscripts for your reference.
- Sex-specific epigenetics drive low GPER expression in gastrointestinal smooth muscles in type 2 diabetic mice (Muhammad A., et.al., 2024,Nature/Scientific Reports).
- Ticagrelor suppresses oxidized low‑density lipoprotein‑induced endothelial cell apoptosis and alleviates atherosclerosis in ApoE‑/‑ mice via downregulation of PCSK9
- MudPIT analysis of released proteins in Pseudomonas aeruginosa laboratory and clinical strains in relation to pro-inflammatory effects
- Treatment With Wheat Root Exudates and Soil Microorganisms From Wheat/Watermelon Companion Cropping Can Induce Watermelon Disease Resistance Against Fusarium oxysporum f. sp. Niveum
- TRPC6 fulfills a calcineurin signaling circuit during pathologic cardiac remodeling
- Optimum salinity for Nile tilapia (Oreochromis niloticus) growth and mRNA transcripts of ion-regulation, inflammatory, stress- and immune-related genes
- Regulation of Atp7a RNA contributes to differentiationdependent Cu redistribution in skeletal muscle cells.
- Resveratrol attenuates acute kidney injury by inhibiting death receptor‑mediated apoptotic pathways in a cisplatin‑induced rat model
- Dual effects of arsenic trioxide on tumor cells and the potential underlying mechanisms.
- Microglia induce neurogenic protein expression in primary cortical cells by stimulating PI3K/AKT intracellular signaling in vitro.
- The GAPDH antibody in Western blot assays.
- T-plastin expression downstream to the calcineurin/NFAT pathway is involved in keratinocyte migration.
- The role of SUMO-conjugating enzyme Ubc9 in the neuroprotection of isoflurane preconditioning against ischemic neuronal injury.
- Hormetic Effect of Berberine Attenuates the Anticancer Activity of Chemotherapeutic Agents.
- In addition, inspecting the raw data excel file provided, indicates that the authors paired the data i.e. NOD mouse A (e.g. MCC-NOD-F1) is normalized to control mouse A (e.g. MCC-CT-F1) and NOD Mouse B (e.g. MCC-NOD-F2) is normalized to control mouse B (e.g. MCC-CT-F2) and so on. What is the basis for pairing of the mice here i.e. why is NOD mouse 2A connected to control mouse A, rather than control mouse B for instance? If this is indeed not substantiated, the individual NOD samples should be normalized to the average of the control values.
Response: The animals were 10-14 weeks old, and sacrificed on different days within the same week in cohorts of 4 animals each (one animal of each sex in each group, that is, 1 CTR male, 1CTR female, 1 NOD male, 1 NOD female). The methods have been updated to reflect this. They are paired based on the day that they were sacrificed and their ages. CTR male animal A was exactly the same age as NOD male animal A, and was sacrificed the same day in the same sitting. Same with CTR female animal A and NOD female animal A. The next time, then, a similar cohort of 4 was sacrificed, and so on, over the course of the study.